# Large Language Models are
# Visual Reasoning Coordinators

**Liangyu Chen**[*,†,♥]  **Bo Li**[*,♥]  **Sheng Shen**[♣]  **Jingkang Yang**[♥]
**Chunyuan Li**[♠]  **Kurt Keutzer**[♣]  **Trevor Darrell**[♣]  **Ziwei Liu**[♥,✉]
[♥] S-Lab, Nanyang Technological University
[♣] University of California, Berkeley   [♠] Microsoft Research, Redmond
{lchen025, libo0013, ziwei.liu}@ntu.edu.sg
https://github.com/cliangyu/Cola

## Abstract

Visual reasoning requires multimodal perception and commonsense cognition of the world. Recently, multiple vision-language models (VLMs) have been proposed with excellent commonsense reasoning ability in various domains. However, how to harness the collective power of these complementary VLMs is rarely explored. Existing methods like ensemble still struggle to aggregate these models with the desired higher-order communications. In this work, we propose 🥤**Cola**, a novel paradigm that coordinates multiple VLMs for visual reasoning. Our key insight is that a large language model (LLM) can efficiently coordinate multiple VLMs by facilitating natural language communication that leverages their distinct and complementary capabilities. Extensive experiments demonstrate that our instruction tuning variant, 🥤**Cola-FT**, achieves state-of-the-art performance on visual question answering (VQA), outside knowledge VQA, visual entailment, and visual spatial reasoning tasks. Moreover, we show that our in-context learning variant, 🥤**Cola-Zero**, exhibits competitive performance in zero and few-shot settings, without finetuning. Through systematic ablation studies and visualizations, we validate that a coordinator LLM indeed comprehends the instruction prompts as well as the separate functionalities of VLMs; it then coordinates them to enable impressive visual reasoning capabilities.

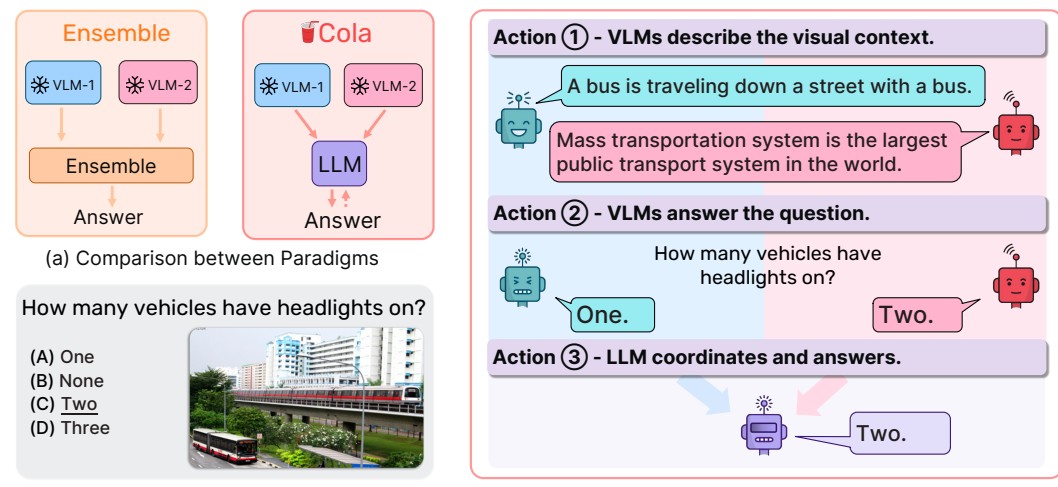

(a) Comparison between Paradigms

(b) Example Visual Reasoning Task: VQA

(c) Cola Solution to the Example VQA

Figure 1: We propose, 🥤Cola, using a coordinative language model for visual reasoning. Cola coordinates multiple pretrained VLMs based on the visual context and plausible answers they provide.

*Equal Contribution.    †Project Lead.    ✉Corresponding Author.

37th Conference on Neural Information Processing Systems (NeurIPS 2023).

# 1 Introduction

**Visual reasoning** is a crucial task that demands models to not only comprehend and interpret visual information but also to apply high-level cognition to derive logical solutions [40, 120, 60]. The field has received significant attention from the machine learning community because of its potential to enable a wide range of intelligent applications, such as intelligent tutoring systems [5, 94, 69], automated image captioning [103], and virtual assistants [88, 50]. To perform visual reasoning effectively, a model must possess both visual perception capabilities and strong logic reasoning abilities.

While classic visual reasoners typically rely on complex architectures [117, 61, 116] or are unable to generalize beyond the training dataset [121, 72], recent advancements in large pretrained models have shown that vision-language models (VLMs) can achieve impressive performance on visual reasoning tasks even under zero-shot settings [104, 52, 51, 49, 48, 3]. Meanwhile, large language models (LLMs) have also demonstrated robust zero-shot commonsense reasoning abilities on the natural language processing (NLP) applications [8, 15, 109]. Several recent studies have attempted to combine such complementary VLMs and LLMs for visual reasoning. For example, PICa [115] utilizes image captioning models to generate textual prompts for GPT-3 [8], and adapts GPT-3 to solve the visual question answering (VQA) tasks in an in-context few-shot learning manner. Socratic Models [123] allow VLMs and LLMs to communicate through prompt engineering to unlock zero-shot multimodal reasoning capabilities.

On the premise that current studies have focused on the interactions among heterogeneous models (specifically, among VLM and LLMs), in this work, we shift to examine how to reconcile **homogeneous expert models** (*e.g.*, multiple VLMs) with an LLM in a *coordinative* paradigm. Inspired by the findings in CICERO [65] that LLMs capture strong strategic planning and negotiation abilities in coordinating multiple agents, we propose 🥤**Cola**, a novel **model ensemble** approach that utilizes an LLM as the coordinator in between multiple VLMs. A key finding of this study is that *given multiple VLMs with different preferred patterns in describing the visual context and predicting plausible answers in natural languages, an LLM can coordinate and integrate their respective strengths efficiently and effectively*. We present two variants of Cola, namely 🥤**Cola-FT** and 🥤**Cola-Zero**, where FT corresponds to an instruction finetuning approach and Zero stands for an in-context learning approach to adapt the coordinator LLM for visual reasoning. Figure 1 provides an overview of Cola and conventional model ensemble approaches.

Existing work on model ensembles usually focuses on manipulating model weights [36] or predictions [111, 22], while remaining cumbersome, if possible, to implement on prevalent end-to-end black box model APIs, like GPT-4 [70], Google Bard, Anthropic Claude, etc. To address this issue, prompt ensembles [107, 106, 73] sample model outputs (*e.g.*, rationales) in natural languages to boost chain-of-thought reasoning [109]. Recent studies on augmented LLM such as [86, 93, 58] have delved into developing a comprehensive strategy that enables LLMs to utilize external tools. These tools comprise multiple off-the-shelf models, web search engines, Python functions [97], and rule-based modules, which are instrumental in performing complex tasks. Despite these efforts, the power of prompt ensembles to aggregate multiple models remains untouched. In contrast, we show that Cola leverages language prompts generated from multiple expert models to make model ensembles.

Systematic experiments demonstrate that Cola performs at the pinnacle of ability on VQA, outside knowledge VQA, visual entailment, and visual spatial reasoning tasks. Specifically, Cola-FT achieves state-of-the-art performance on A-OKVQA [89], OK-VQA [63], e-SNLI-VE [21], and VSR datasets [56], even when compared with methods that adopt larger models or require more training computations. Cola-FT also demonstrates competitive capabilities on VQA v2 [27], and compositional reasoning tasks (GQA [35] and CLEVR [40]). Perhaps surprisingly, we find that Cola-Zero demonstrates comparable performance without finetuning, as an emerging ability of larger language models. Compared to a single VLM and ensemble modeling, both Cola-FT and Cola-Zero improve the performance substantially across most datasets. They are even effective with the recent large multimodal models like InstructBLIP [18] which embeds a pretrained LLM inside itself. Besides, we conduct a thorough analysis of perturbed VLM caption or plausible answer labels and saliency visualization to investigate how Cola recognizes each VLM's individual functionalities and then performs coordination behaviors. We conjecture that, in principle, any language-expressing reasoning task can be usefully augmented with coordinative language models that learn to aggregate multiple expert models, even via in-context learning.

In summary, our contributions are as follows:

- **Cola**: a novel paradigm that utilizes a language model as a coordinator between multiple vision-language models to integrate their respective strengths for visual reasoning (§2).
- **State-of-the-art performance**: Cola achieves the pinnacles on a challenging suite of diverse visual reasoning tasks and datasets(§3.2).
- **Systematic analysis**: our experiments reveal how Cola comprehends the instruction prompts, then coordinates them to capture impressive visual reasoning capabilities (§3.3, §3.4, §3.6).

## 2 🥤Cola

We formulate various visual reasoning tasks as a multi-class classification problem for simplicity. Given an image $v \in \mathcal{V}$ and a question-like prompt $q \in \mathcal{Q}$, the reasoner is required to select an answer $a$ from the candidate set $\mathcal{A} = \{a\}$. In the case that the reasoner outputs a text sequence $s_{v,q}$, we map $s$ to a prediction $P(v,q) = \mathrm{sim}(T(s_{v,q}), T(\{a\}))$ where $T$ transforms text sequences into text embeddings (we use a `all-mpnet-base-v2` model [81] here), and sim denotes cosine similarity.

**Ensemble Modeling**   aggregates multiple models' predictions in order to improve the overall performance (Figure 1a). For instance, one common practice is averaging over $n$ models:

$$P(v,q) = \frac{1}{n} \sum_{i=1}^{n} P_i(v,q), \tag{1}$$

where $P_i(v,q)$ denotes the prediction of the $i^{th}$ model on input $(v,q)$.

### 2.1 🥤Cola & Templates

An overview of Cola is shown in Figure 1c. We use OFA [104] and BLIP [52] as the VLMs. LLMs include encoder-decoder (FLAN-T5 [16]) and decoder-only (Vicuna-1.5 [125], Mistral [39]) transformers. We first prompt each VLM to output captions and plausible answers independently. We then concatenate the instruction prompt, the question with choices, captions, and plausible answers to fuse all contexts for the LLM to reason, coordinate, and answer.

| **General Prompt Template** |
|---|
| Answer the following multiple-choice question by OFA and BLIP's description and their answers to the visual question. OFA and BLIP are two different vision-language models to provide clues. OFA's description: `<OFA caption>` BLIP's description: `<BLIP caption>` Q: `<Question>` OFA's answer: `<OFA answer>` BLIP's answer: `<BLIP answer>` Choices: `<Choices to the question>` A: |

Table 1: **LM prompt template.** The LM is instructed to coordinate VLMs. Each question set defines *visual context*, *question (and choices)*, and *plausible answers*.

**Image captioning**   gives important visual context to reason from. We first employ $i^{th}$ VLM to describe each image respectively to get visual descriptions $c_i(v)$. We use `ofa-large` for OFA and `blip-image-captioning-large` for BLIP, both implemented by the Hugging Face Transformers library [110].

**Plausible answers**   by the VLMs to the question provide clues and patterns of VLMs for the LM to consider and coordinate. Similar to captioning, we prompt each $i^{th}$ VLM using the image-question pair to get a plausible answer $\hat{a}_i(v,q)$. We use `ofa-large` for OFA and `blip-vqa-base` for BLIP. Following OFA, our prompt template varies by task category. For the VQA tasks, we leave the original question unchanged. For the visual entailment and visual spatial reasoning tasks, our prompt template is *" does the image describe "`<text premise>`" ?".*

**Prompt template**   is shown in Table 1. First, we designed an instruction prompt for LM to understand the requirement to coordinate VLMs to answer the visual reasoning question. We then concatenate the captions from each VLM model, with the VLM identification labels in natural languages (referred to as VLM caption labels in Figure 3), such as *"OFA's description: `<OFA caption>`"*. Next, the question and its plausible answers provided by VLMs (with similar identification labels referred to as VLM answer labels in Figure 3) are concatenated. We follow [16] to

include the choices of question (for multiple-choice questions in A-OKVQA, e-SNLI-VE, and VSR) and *"A:"* to prompt for answers. Overall, the prompt for LLM input is given by:

$$\text{Prompt}(v, q) = \text{Template}(\{(c_i(v), \hat{a}_i(v, q)) \mid i = 1, \cdots, n\}). \tag{2}$$

More specific prompt templates on each dataset are provided in Appendix A.7.

## 2.2 🥤 Cola-FT

**Instruction Tuning**  of Cola is initialized with pretrained checkpoints. Given the question $q$ based on the image $v$, the LM predicts the answer in the form of sequence

$$s_{v,q} = \text{LLM}(\text{Prompt}(v, q)). \tag{3}$$

To optimize the LLM, we use the language modeling loss for next-token prediction, with the teacher forcing strategy. We only finetune the LLM (while not the VLMs) to follow the common paradigm of ensemble modeling and simplify the method (Figure 1).

**Inference**  deploys the same prompt as Table 1 to align with instruction tuning. We resort to the greedy decoding strategy for conditional sequence generation at both instruction tuning and inference.

## 2.3 🥤 Cola-Zero

**In-context learning**  is an emerging ability of the LLM models pretrained on documents of long-range coherence. By learning input and output format from demonstration, in-context learners learn to perform a downstream task simply by conditioning on a prompt consisting of input-output examples [114]. The coordinator LLM, finetuned on instruction prompts with examples, is capable of in-context few-shot learning and zero-shot learning (see Figures 6 and 7).

**Cola-Zero**  is the in-context few-shot/zero-shot learning variant of Cola, without instruction tuning. For in-context $k$-shot learning, we modify the prompt (Table 1) to include $k$ input-output examples sampled from the training set. For zero-shot learning, the prompt remains the same as Table 1.

## 3   Experiments

First, the experimental setups and basic methods are described in this section. The main quantitative results are then presented in Table 2. Next, we analyze qualitative visualizations and scaling to verify the effectiveness of the Cola paradigm in different settings. Further details on datasets, training, evaluation, and experimental analysis can be found in Appendix A.

### 3.1   Baseline Methods

**State-of-the-art Methods**  are in two broad categories, VLM alone, and VLM combined with LLM. In Table 2, for a fair comparison, we detail the techniques (whether finetuning or in-context learning is required) used for training VLMs and LLMs, and the number of training epochs.

**Ensemble Modeling**  can be considered the most basic baseline for aggregating VLMs. It represents the base performance that the combination of VLMs can achieve on the target task when not trained. We implement an averaging ensemble (Equation (1)) of cosine similarity between VLM output and each choice of a question as our ensemble baseline.

### 3.2   Overall Performance

In Table 2, we first observe that Cola-FT achieves state-of-the-art (SOTA) performance on four datasets (A-OKVQA, OK-VQA, e-SNLI-VE, VSR), with merely 1 epoch of instruction tuning and a medium-sized language model. In contrast, many previous SOTA methods require finetuning more epochs than Cola-FT (*e.g.*, VLC-BERT, PromptCap on A-OKVQA). Some also use much larger language models, such as GPT-3 (175B) [8] and OPT (175B) [124]. Cola-FT outperforms OFA-X on e-SNLI-VE, although the latter is finetuned on much more related tasks and data (c.f.

Table 2: **Overall performance.** Model Spec. denotes specification where we summarize the detailed VLMs and LMs adopted in each method and their parameters. FT and ICT denote finetuning and in-context learning, respectively. Downward arrows indicate that fewer FT and ICT are more efficient. The accuracy metric varies slightly in different datasets. In A-OKVQA, we report both val/test accuracies, and val accuracy in VQA v2, OK-VQA, e-SNLI-VE, GQA, and CLEVR; test (zero-shot split) accuracy in VSR. Upward arrows indicate higher accuracy is better. We mark the best performance on each dataset with **bold font** and second-best with underlines.

| Method | Vision-language Model | | Large Language Model | | | Accuracy ↑ |
|---|---|---|---|---|---|---|
| | Model Spec. | FT↓ | Model Spec. | ICL↓ | FT↓ | |
| **Visual Question Answering (VQA v2)** | | | | | | |
| MetaLM [31] | Pretrained Encoder | 350k steps | MetaLM (1.3B) | - | 350k steps | 41.1 |
| PNP-VQA [99] | BLIP-Caption (446M) | - | UnifiedQAv2 [43] (11B) | 0-shot | - | 63.3 |
| BLIP-2 [51] | CLIP [76] (1.2B trainable) | 5 epochs | FLAN-T5 (3B) | - | - | 81.6 |
| BLIP-2 [51] | CLIP [76] (1.2B trainable) | 5 epochs | OPT [124] (6.7B) | - | - | 82.2 |
| Ensemble | | - | - | - | - | 68.0 |
| **Cola-Zero** | BLIP+OFA (384M+472M) | - | FLAN-T5 (11B) | 2-shot | - | 69.1 |
| **Cola-FT** | | - | FLAN-T5 (11B) | - | 1 epoch | **83.7** |
| **Outside Knowledge Visual Question Answering, Multiple Choice (A-OKVQA)** | | | | | | |
| PromptCap [32] | OFA (472M) | 2 epochs | GPT-3 (175B) | 0-shot | - | - / 73.2 |
| Img2Prompt [29] | BLIP (384M) | - | OPT (175B) | 0-shot | - | 42.9 / 40.7 |
| Prophet-MC [90] | MCAN-large [118] (56M) | 6 epochs | GPT-3 (175B) | 16-shot | - | 76.4 / 73.6 |
| Ensemble | | - | - | - | - | 56.6 / 54.9 |
| **Cola-Zero** | BLIP+OFA (384M+472M) | - | FLAN-T5 (11B) | 0-shot | - | 65.4 / 61.6 |
| **Cola-Zero** | | - | FLAN-T5 (11B) | 2-shot | - | 70.4 / 66.5 |
| **Cola-FT** | | - | FLAN-T5 (11B) | - | 1 epoch | 77.7 / 74.0 |
| **Cola-Zero** | | - | FLAN-T5 (11B) | 0-shot | - | 68.0 / 66.5 |
| **Cola-Zero** | | - | FLAN-T5 (11B) | 2-shot | - | 72.3 / 72.3 |
| **Cola-FT** | InstructBLIP [18] | - | FLAN-T5 (11B) | - | 1 epoch | **78.1** / **76.7** |
| **Cola-Zero** | XL+XXL | - | Vicuna (7B) | 2-shot | - | 63.9 / 63.0 |
| **Cola-FT** | (3B+11B) | - | Vicuna (7B) | - | 1 epoch | 68.6 / 66.9 |
| **Cola-Zero** | | - | Mistral (7B) | 2-shot | - | 69.3 / 66.2 |
| **Cola-FT** | | - | Mistral (7B) | - | 1 epoch | 74.3 / 71.8 |
| **Outside Knowledge Visual Question Answering, Direct Answer (OK-VQA)** | | | | | | |
| PromptCap [32] | OFA (472M) | 2 epochs | GPT-3 (175B) | 0-shot | - | 58.8 |
| Prophet [90] | MCAN-large [118] (56M) | 6 epochs | GPT-3 (175B) | 16-shot | - | 61.1 |
| Ensemble | | - | - | - | - | 39.2 |
| **Cola-Zero** | BLIP+OFA (384M+472M) | - | FLAN-T5 (11B) | 0-shot | - | 39.4 |
| **Cola-Zero** | | - | FLAN-T5 (11B) | 2-shot | - | 39.4 |
| **Cola-FT** | | - | FLAN-T5 (11B) | - | 1 epoch | **62.4** |
| **Visual Entailment (e-SNLI-VE)** | | | | | | |
| e-UG [42] | UNITE (86M) | 400 epochs | GPT-2 (117M) | - | 400 epochs | 79.5 |
| OFA-X [74] | OFA (472M) | 10 epochs | - | - | - | 80.9 |
| Ensemble | | - | - | - | - | 48.8 |
| **Cola-Zero** | BLIP+OFA (384M+472M) | - | FLAN-T5 (11B) | 0-shot | - | 56.2 |
| **Cola-Zero** | | - | FLAN-T5 (11B) | 2-shot | - | 57.8 |
| **Cola-FT** | | - | FLAN-T5 (11B) | - | 1 epoch | **81.6** |
| **Visual Spatial Reasoning (VSR)** | | | | | | |
| VisualBERT [53] | VisualBERT (110M) | 100 epochs | - | - | - | 54.0 |
| LXMERT [98] | LXMERT (110M) | 100 epochs | - | - | - | 63.2 |
| ViLT [44] | ViLT (88M) | 30 epochs | - | - | - | 62.4 |
| Ensemble | | - | - | - | - | 51.4 |
| **Cola-Zero** | BLIP+OFA (384M+472M) | - | FLAN-T5 (11B) | 0-shot | - | 55.8 |
| **Cola-Zero** | | - | FLAN-T5 (11B) | 2-shot | - | 54.9 |
| **Cola-FT** | | - | FLAN-T5 (11B) | - | 1 epoch | **67.0** |
| **Compositional Question Answering, Real Images (GQA)** | | | | | | |
| BLIP [52] | BLIP (384M) | - | - | - | - | 41.7 |
| OFA [104] | OFA (472M) | - | - | - | - | 58.0 |
| VisProg [30] | ViLT (88M) | - | GPT-3 (175B) | 8-shot | - | 50.5 |
| **Cola-FT** | BLIP+OFA (384M+472M) | - | FLAN-T5 (11B) | - | 1 epoch | **60.3** |
| **Compositional Question Answering, Synthetic Images (CLEVR)** | | | | | | |
| InstructBLIP [18] | XL (3B) | - | - | - | - | 33.7 |
| | XXL (11B) | - | - | - | - | 16.6 |
| **Cola-Zero** | InstructBLIP | - | FLAN-T5 (11B) | 2-shot | - | 34.4 |
| **Cola-FT** | XL+XXL (3B+11B) | - | FLAN-T5 (11B) | - | 1 epoch | **54.3** |

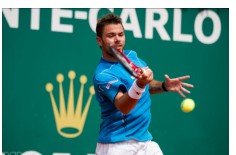 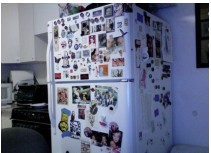 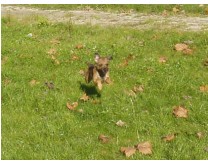 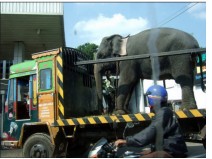

| | | | | |
|---|---|---|---|---|
| Question | What type of shot is the man hitting? | What appliance is next to an appliance that is highly decorated? | Does this image describe "puppy running after a stick in grass" ? | Does this image describe "The truck is away from the elephant" ? |
| OFA caption | tennis player hits a return to tennis player during their men's singles second round match at | a refrigerator covered in a variety of stickers. | a coyote is seen in this undated file photo. (credit: ktla | an elephant is loaded onto a truck in yangon. photo: afp |
| BLIP caption | a man in a blue shirt is playing tennis | a refrigerator with many pictures on it | a dog running through the grass in a field | a man riding a motorcycle with a truck behind him |
| Choices | ['forehand', 'backhand', 'serve', 'dropshot'] | ['mixer', 'stove/oven', 'refrigerator', 'microwave'] | ['yes', 'maybe', 'no'] | ['yes', 'no'] |
| OFA answer | backhand | stove/oven | yes | yes |
| BLIP answer | forehand | microwave | no | no |
| Cola-Zero answer | forehand | stove/oven | no | no |
| Cola-FT answer | forehand | stove/oven | maybe | no |
| Cola-FT answer (swapped VLM answer labels) | backhand | microwave | maybe | yes |

Figure 2: **Qualitative examples.** The correct choices are underlined. Leftmost: a commonsense question example of A-OKVQA; LLM follows the answer of BLIP. Left: a visual question example of A-OKVQA; LLM follows the answer of OFA. Right: an example of e-SNLI-VE; LLM chooses another option after coordination. Rightmost: an example of VSR; LLM predicts based on the caption of OFA and the answer of BLIP. Cola-Zero answers are inferenced in zero-shot settings. The bottom row, Cola-FT (swapped VLM answer labels), indicates that the LLM follows the answer of certain VLMs based on their separate functionalities. LLM answers are associated with the distribution of VLM answer labels.

Cola-FT is trained on each one dataset only in Table 2). In addition, the lighter variant Cola-Zero also achieves comparable performance to most baseline methods through in-context few-shot and zero-shot learning, without training any model parameter. To evaluate the performance of Cola with large multimodal models that embed large language models inside, we also assembled InstructBLIP [18] models based on FLAN-T5 XL and XXL and tested on A-OKVQA dataset. In Table 2, we report the multiple-choice accuracies on A-OKVQA. See Appendix A.4 for direct answer results.

## 3.3 Qualitative Examples

In Figure 2, we exhibit several qualitative examples. The language coordinator determines the correctness of VLM plausible answers implicitly, given their captions and the caption and answer labels. The leftmost example (a tennis player playing) demonstrates a case when captions are not informative to guide the LLM for predictions. Between OFA and BLIP's plausible answers, the LLM follows the answer of BLIP. In contrast, in the left example (an oven next to a fridge), again with trivial captions, the LLM follows OFA's plausible answer instead.

It's all plausible answers, captions, VLM answer/caption labels, and the world knowledge the LLM encodes in itself, that contribute to the final decision of the language coordinator. The rightmost example presents the scenario of inconsistency between captions and answers. OFA describes the image as *"an elephant is loaded onto a truck in yangon."* Though, it agrees that *"the truck is away from the elephant"*. With Cola-FT, The LLM coordinates OFA's correct caption and BLIP's correct answer to make a reasonable prediction.

Notably, we observe a scenario in which captions can be more informative than plausible answers to guide LLM. The right example (a puppy running) presents an uninformative image. Though neither OFA nor BLIP succeeds in answering the question, the LLM chooses to answer with "maybe" based on the given visual context. See Appendix A.5 and Appendix A.6 for more analysis on qualitative examples and failure cases.

## 3.4 Coordination Analysis

Overall, Figure 3 validates the efficacy of Cola to coordinate VLMs. All the experiments use the same prompt template as in Table 1 unless otherwise stated. On A-OKVQA validation set, the performance of a single VLM (w/o FLAN-T5) is 50.83% for BLIP, or 54.75% for OFA. To validate the effectiveness of multi-VLM collaboration, we first ablate single-VLM variants of Cola-FT, shown as #1 (OFA only, without BLIP) and #2 (OFA only, without OFA) from the top. As expected, both fall behind Cola-FT (#8) by a large margin. With both VLMs, we ablate VLMs' captions (#3) and VLMs' plausible answers (#4), which reveal that plausible answers are much more significant in helping the LLM answer visual reasoning questions. Next, we perturb caption labels by swapping the VLM caption labels at instruction tuning and evaluation (#5), specifically *"OFA's description: "* and *"BLIP's description: "*, by a chance of 50%. Under such settings, the LLM fails to acquire the preferred patterns of VLM for

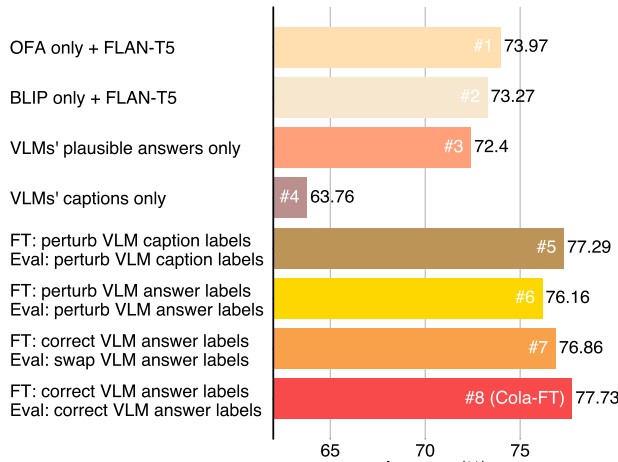

Figure 3: **Ablation study results** using a single VLM (#1, #2 from top), VLMs' plausible answers only (w/o captions, #3), VLMs' captions (w/o plausible answers, #4), perturbed VLM caption/answer labels at instruction tuning (#5, #6), and swapped answer labels at evaluation (#7). In #6, the coordination prior cannot be learned by the LLM. In #7, the coordination prior can be learned by the LLM, but cannot be properly applied at evaluation. FT: instruction tuning; Eval: evaluation.

captioning, though the overall visual context is preserved. The results underperform Cola-FT, which verifies that VLM caption labels improve Cola-FT performance. Notably, the VLM (plausible) answer labels are more important to the LLM's decision: a considerable gap exists between (#6) and Cola-FT. In #6, the LLM fails to learn the separate functionalities of VLM when answer labels are perturbed. This highlights that the performance gained from the *coordination* between BLIP and OFA, but not the strong reasoning capabilities of the LLM, FLAN-T5.

Naturally, we ask *what if the LLM can learn the patterns each VLM answers, but they cannot apply it at inference?* We input correct VLM answer labels at instruction tuning and swap labels at evaluation (#7). Consequently, #7 falls behind Cola-FT with a smaller but still considerable margin. The results suggest that learning and applying the separate functionalities of VLMs is important for the coordinator LLM to make predictions. See Appendix A.9 for more ablation studies.

## 3.5 Scaling Cola

| Methods | A-OKVQA | e-SNLI-VE |
|---|---|---|
| OFA-base (1) | 45.76 | 52.60 |
| OFA-base (2) | 46.07 | 51.70 |
| OFA-base (3) | 45.73 | 52.33 |
| Ensemble (majority voting) | 44.79 | 52.71 |
| Ensemble (average) | 46.04 | 52.25 |
| **Cola-Zero** (2-shot) | 47.71 | 54.42 |
| **Cola-FT** | 48.85 | 56.92 |

Table 3: **Performance of ensemble methods based on three identical models.**

| Methods | A-OKVQA | e-SNLI-VE |
|---|---|---|
| OFA-tiny | 39.03 | 50.20 |
| OFA-medium | 42.45 | 51.04 |
| OFA-base | 45.76 | 52.60 |
| Ensemble (majority voting) | 46.71 | 53.94 |
| Ensemble (average) | 46.62 | 54.41 |
| **Cola-Zero** (2-shot) | 49.37 | 57.63 |
| **Cola-FT** | 54.26 | 63.68 |

Table 4: **Performance of ensemble methods based on three different models.**

**Scaling Cola with More VLMs.** By decoding the top-k (k=5) results from three identical (OFA-base) models on the A-OKVQA validation set, both the answers and captions may exhibit slight variations. Cola demonstrated significant performance improvements compared to a single VLM or ensemble, as shown in Table 3. Furthermore, the performance gap between the ensemble baselines and Cola based on three different models (OFA-tiny/medium/base) is even more substantial, as depicted in Table 4.

We further scale the number of OFA-base and find that Cola-FT satures at 5 VLMs (49.77%) and Cola-Zero (2-shot in-context learning) satures at 3 VLMs (47.71%). We observe that long input harms the performance of Cola-Zero, which is negative for scaling with more VLMs (Figure 4). LMs with a larger context window size [12, 68, 9] are promising to further improve the performance of Cola, which we leave for future works.

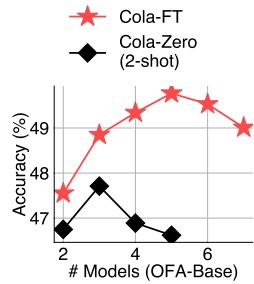

Figure 4: **Scaling of # Models.**

**Scaling Model Size.** We conduct experiments on scaling the coordinator LLM size to see if there are ramifications when operating at a larger scale. Figure 5 reveals that Cola-FT performance increases as the LLM (FLAN-T5) model size increases. Notably, Cola-FT/small, with only 80M parameters, could achieve 65% MC accuracy on A-OKVQA validation set, which is far beyond our baseline methods (55%). Cola-Zero, under the in-context learning paradigm, achieves competitive performance when the model grows to a billion-parameter scale. This observation on Cola-Zero can be regarded as a proof-of-concept that potentially reveals Cola-Zero's emerging abilities (inherited from FLAN-T5 [16]) on visual reasoning tasks at a relatively large scale. Cola-FT is effective with small models, but Cola-Zero is an emerging ability on larger models only.

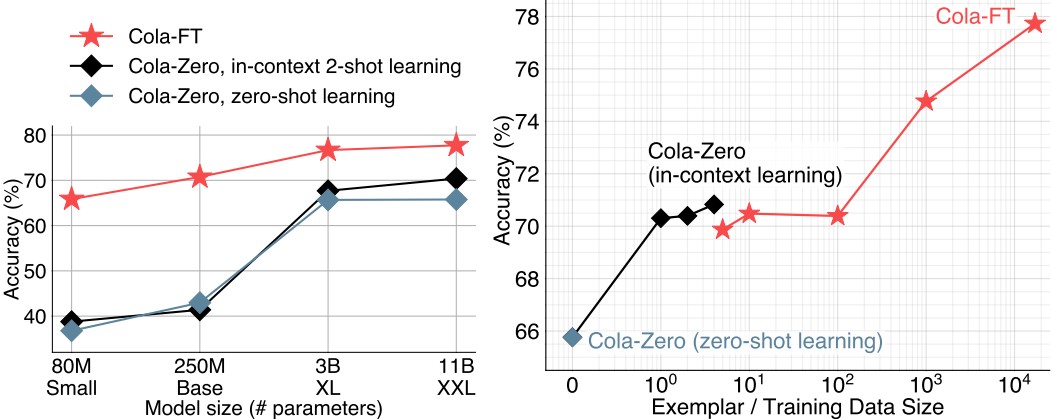

Figure 5: **Cola performances versus the LLM (FLAN-T5) sizes.**

Figure 6: **Low-data Cola-FT and Cola-Zero performances.** $x$-axis is distorted for optimal display.

**In-context Learning & Low-data Instruction Tuning.** We conduct experiments on different data scales to verify Cola's performance varying from zero-shot to full-shot under in-context learning and full-finetune paradigm. As shown in Figure 6, with Cola-Zero, few-shot exemplars substantially improve performance compared to zero-shot learning. As [16, 108] revealed, exemplars potentially help the model better understand the output format and understand the instructions in Table 1. Cola-Zero for in-context few-shot learning outperforms zero-shot learning by a large margin, being on par with low-data Cola-FT without instruction tuning. We also observe Cola-FT's substantial performance gain when finetuning shots increase to 1000 and beyond.

### 3.6 Saliency Visualization

As shown in Figure 7, we visualize the importance of the input prompt tokens by input-gradient saliency feature attribution [19], implementing with `Ecco` [2]. The input tokens that are more relevant to predict the output token `"grass"` are highlighted in darker colors. In the given example, both Cola-FT and Cola-Zero predict the correct answer and find the relevant clues from visual context and plausible answers. Figure 7(b) shows that Cola-Zero attributes the output more to the instructions in the prompt template. This explains Cola-Zero's competitive performance, a consequence of FLAN instruction tuning [108]. After instruction tuning, Cola-FT focuses more on the most informative parts of input: the question, choices, as well as VLMs' plausible answers.

### 3.7 Can 🥤 Cola-FT Explain its Answers?

We modify the prompt template of Cola so that the model would output the logical conduction process, allowing us to observe the specific behaviors of the LLM during its coordination of VLMs.

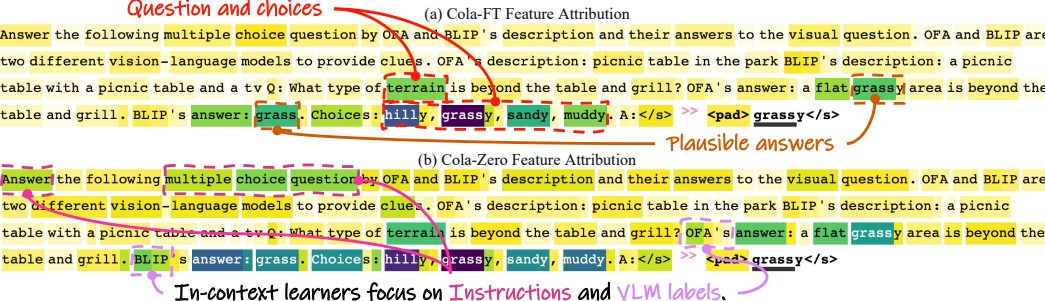

Figure 7: **Visualization of input token saliency.** We visualize the relevancy between input tokens and the output token `"grass"` by feature attribution [19]. The more salient tokens are highlighted in darker boxes. Cola-FT focuses on the question, choices, and VLMs' plausible answers in (a). While as shown in (b), Cola-Zero pays extra attention to instructions and VLM labels, as a consequence of FLAN-T5 instruction tuning [16].

We finetune Cola-FT to output rationales before answers, by A-OKVQA ground truth rationales. In the modified prompt, we ask the model to provide rationales. For the leftmost example of Figure 8, Cola-FT outputs *"Rationale: People might sit here to rest. The umbrellas are on the riverwalk. The answer is:rest "*. OFA gives a reasonable answer (but out of choices) to the question while BLIP gives an irrelevant answer. In this case, both answers are wrong. However, either Cola-Zero or Cola-FT is able to infer from captions and plausible answers to give the correct answer "to rest". The rationale suggests that the LLM understands the scene that the umbrellas are on the riverwalk and guesses that people might sit here to rest based on commonsense. The final answer is correct. For the leftmost example of Figure 11, the rationale output is *"The bike is parked in a no parking zone. The bike is parked next to a pedestrian crossing sign. The answer is:no parking"*. Both VLMs are wrong in their plausible answers. OFA's answer "boating" is semantically correct as the correct answer is "kayaking", though it's not the correct answer because this is a multiple-choice question. Cola-Zero gives a wrong answer "OFA" which is obviously wrong because "OFA" is the name of one of the VLMs given in the prompt and it's out of the choices too. However, Cola-FT gives the correct answer "kayaking", recognizing the correct choice based on prompts of captions and plausible answers after being finetuned. Even though the OFA and BLIP captions fail to identify that the people in the water are on a canoe. The LLM identifies that the people in the water are associated with the canoe. The rationale is valid and helpful, though repetitive. The final answer is correct.

To force the LLM to output rationale does not improve the reasoning performance of Cola (w/t rationale 74.3% vs. w/o rationale 77.7%, on A-OKVQA val set). This might be attributed to the low-quality ground truth rationales provided by the A-OKVQA dataset that we use to train the LLM. Such rationales are just short and objective descriptions of the scene, without suggesting the underlying outside knowledge to answer the question. Therefore, training the LLM to output rationale is harmful, though it derives insights into the LLM's behaviors during reasoning.

### 3.8 Does 🥤 Cola-FT Transfer across Tasks?

We examine Cola-FT's generalization ability across tasks. From Table 5, we observe the zero-shot performances on target datasets after instruction tuning on a certain source dataset. Although each dataset varies in question types and prompt templates (see detailed comparisons in Appendix A.7), we find that Cola-FT maintains competitive performance when zero-shot transferred to a new task, outperforming Cola-Zero in-context 2-shot learning and ensemble baselines (see also Table 2).

Table 5: **Cola-FT is generalizable across out-of-distribution tasks.** In most cases, the performances surpass Cola-Zero (in-context 2-shot learning results in brackets) on target datasets.

| Finetuned on | A-OKVQA val | e-SNLI-VE val | VSR test |
|---|---|---|---|
| A-OKVQA | 77.7 (70.4) | 58.7 | 57.6 |
| e-SNLI-VE | 71.2 | 81.6 (57.8) | 51.7 |
| VSR | 71.4 | 61.4 | 66.9 (55.8) |

## 4   Related Work

**Visual Reasoning.** Beyond unimodal reasoning tasks such as question answering (QA) [100, 14, 119, 7], visual reasoning extends high-level cognition to visual domains, requiring an intelligent agent to derive rational solutions [40, 35, 84, 120, 38, 112]. Several tasks have been introduced to address

visual reasoning, such as VQA [1], in which models are expected to provide answers to questions related to an image, and visual entailment [113], where the model is required to determine if a text description is consistent with the visual content provided.

Classic visual reasoning methods employ an image encoder along with a reasoning block that utilizes attention mechanisms [102, 121, 122, 105], neuro-symbolic methods [101, 117, 61], or external knowledge [62, 28, 13].

Recent progress in large pretrained models has led to the development of LLMs that capture exceptional commonsense reasoning capabilities [78, 16, 15]. These LLMs can potentially replace the reasoning module in visual reasoning tasks, and LLMs' lack of perception can be compensated by incorporating multiple VLMs trained on different domains [76, 104, 52]. However, there is still a lack of research on how to harness the collective power of these separate VLMs for visual reasoning tasks. More related works are in Appendix B.

**Model Ensemble.** Model ensemble is a powerful machine learning technique that combines the predictions of multiple models to improve the overall performance of a given task [20]. The variance and bias of the final predictions decrease, resulting in a more robust and accurate model [83]. To this end, common methods include averaging [111], voting [37], interpolation [36], weighting the predictions based on model performance [22], or stacking the models [10].

Ensemble methods have been challenging for *generative* tasks like visual reasoning, where a simple combination is not applicable to heterogeneous models due to their enormous and varying input/output token spaces. To address the issue, Socratic Models (SMs) [123] use prompt engineering to guide the heterogeneous pretrained multimodal models through natural language discussions. With a similar goal, [54] proposes a closed-loop iterative consensus optimization method to utilize the strengths of individual models. However, previous methods do not fully adapt to the intrinsic patterns of different models, particularly in the visual reasoning scenario. Recent studies, such as CICERO [65], have shown that LLMs possess strong social intelligence in coordinating multiple agents, which inspires us to reorganize pretrained mixed-modal models with a focus on adapting LLMs. More recently, Toolformer [86] and HuggingGPT [93] further demonstrate LLMs' abilities to leverage, coordinate, and incorporate the results from external sources such as other models or even APIs to solve complex tasks. While the external tools are called in sequential order in existing work, we study coordinating multiple tools (specifically, expert models) in parallel in this work.

## 5 Discussion

**Question Format.** Datasets like VQA v2 and OK-VQA contain open-ended questions, while A-OKVQA, e-SNLI-VE, and VSR use multiple-choice. Converting VQA v2 and OK-VQA to classification introduces complexities for traditional ensemble methods, as evident in Table 2. Classic methods struggle with generative models like API-based GPT-4, underscoring Cola's value as an end-to-end ensemble strategy for extensive (vision-)language models. Moreover, Cola-Zero's efficiency also relies on the question format – it's easier for LLMs to answer when given choices like in A-OKVQA. Conversely, Cola-FT finetunes LLMs to discern answer formats (Figure 7).

**Limitations.** Visual reasoning is a diverse topic. This work demonstrates the first step toward applying end-to-end language models for visual reasoning. While the LLMs perform well on the discussed datasets, there is a large body of visual reasoning tasks to evaluate in future works, such as intention prediction and rationale explanation.

**Future Works.** First, exploring the use of non-parametric tools for visual reasoning would be useful to enhance Cola's performance. Second, Cola's use can be extended to other reasoning and planning tasks, such as image generation and action planning, by coordinating multiple models in parallel. Third, by improving inter-model communications, Cola can be more interpretable and safe for high-stakes applications.

**Conclusion.** In this paper, we have proposed a novel paradigm for visual reasoning that harnesses the power of multiple VLMs by utilizing a coordination mechanism, where an LLM acts as a coordinator who communicates with VLMs to integrate their respective strengths. Experiments show that reasoning performance is substantially improved by LLM finetuning or in-context learning. Our results provide a promising step towards building multi-component intelligent systems that capture multimodal reasoning capabilities in a human-like way.

## Acknowledgements

This research/project is supported by the National Research Foundation, Singapore under its AI Singapore Programme (AISG Award No: AISG2-PhD-2022-01-029). Besides, this project is supported by NTU NAP, MOE AcRF Tier 2 (MOE-T2EP20221-0012), and under the RIE2020 Industry Alignment Fund – Industry Collaboration Projects (IAF-ICP) Funding Initiative, as well as advise from the industry partner(s).

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

# A Experimental Details

In this section, we elaborate on our training and evaluation details, prompt templates, and more qualitative examples for analysis.

## A.1 Datasets

Our experiments are conducted on a challenging suite of three diverse visual reasoning tasks, including outside knowledge VQA, visual entailment, and visual spatial reasoning. For each task, we select the following dataset respectively.

**Visual Question Answering v2** [27] (VQA v2) is a large-scale benchmark containing over 1 million images from the COCO dataset and more than 250,000 human-generated question-answer pairs. The dataset is designed to test the ability of machine learning models to understand both the visual content of an image and the meaning behind natural language questions. The questions in VQA v2 cover a wide range of topics and are often open-ended, requiring models to reason and generalize about the world. VQA v2 has been widely used to evaluate the performance of state-of-the-art models in the field of computer vision and natural language processing.

**Augmented Outside Knowledge VQA** [89] (A-OKVQA) contains about 25k questions paired with both multiple choice (MC) answer options. Unlike most existing VQA datasets, the questions in A-OKVQA cannot often be answered by querying the knowledge base, but rather involve some type of commonsense reasoning and outside knowledge about the situation portrayed in the image.

**Outside Knowledge VQA** [63] (OK-VQA) includes more than 14,000 questions that require external knowledge to answer. The answers are provided in free-text direct answer form. Both A-OKVQA and OK-VQA sample images from the COCO dataset, with no overlapping.

**e-SNLI-VE** [21] dataset is an extended version of SNLI-VE dataset [113], which contains about 190k question pairs and human-annotated natural language explanations for the ground-truth labels. The text premise provides a statement about the contents of the image. The task is to determine whether the statement is true or false based on the image content.

**Visual Spatial Reasoning** [56] (VSR) consists of 65 spatial relations (*e.g.,* under, in front of, facing, *etc.*) of instances in images. VSR has more than 10k question pairs, associated with 6940 images from MS COCO [55].

**GQA** [35] dataset consists of 22M questions about various day-to-day images. The questions are about compositional question answering based on scene graphs. In our evaluation, we only use the text of the question as model input, but not the scene graphs.

**Compositional Language and Elementary Visual Reasoning** [40] (CLEVR) is a synthetic dataset with questions that test various aspects of visual reasoning including attribute identification, counting, comparison, spatial relationships, and logical operations. The dataset contains 700k questions in the training set and 150k in the validation set.

## A.2 Instruction Tuning Details

We adopt pretrained BLIP [52][1] and OFA [104][2] as VLMs unless specified otherwise, and freeze their parameters without updating. The instruction tuning only happens on the language model part. The training set of each dataset is used for finetuning. We use the whole training set unless otherwise specified in the low-data instruction tuning discussion.

We use an AdaFactor optimizer [92] at the learning rate of 1e-4 for all Cola-FT experiments. The batch size is by default set to 16, though we find Cola-FT insensitive to batch size. We finetune and evaluate the models on NVIDIA V100 or A100 GPUs. The finetuning time is shown in Table 6.

---

[1]BLIP: https://github.com/salesforce/BLIP

[2]OFA: https://huggingface.co/OFA-Sys

Following the common experiment protocols, we employ a teacher forcing and greedy decoding strategy for fine-tuning.

| V100 hours | A-OKVQA | e-SNLI-VE | VSR | GQA | VQA v2 | OK-VQA | CLEVR |
|---|---|---|---|---|---|---|---|
| Cola-FT | 12 | 8 | 8 | 24 | 80 | 12 | 24 |

Table 6: **Cola-FT training time of FLAN-T5-XXL for each dataset.** We finetune a subset of GQA.

## A.3   Evaluation Details

As specified, we use the validation or test set multiple choice accuracy as the evaluation metric. In A-OKVQA, we report `val/test` accuracy, and `val` accuracy in e-SNLI-VE, `test` (zero-shot split) accuracy in VSR. For simplicity and consistency, we evaluate ablation experiments on A-OKVQA validation set. Following the common experiment protocols [32, 74], we report the single run results for performance comparison.

The exemplars at the inference of Cola-Zero are randomly sampled from the training set, i.e., supposedly help the LLM learn the input data distribution and output format but do not leak relevant information to the evaluation question.

## A.4   A-OKVQA Direct Answer Results

In addition to MC accuracy, we present the direct answer (DA) accuracy of models on the A-OKVQA validation set in Tables 7 and 8.

| | FLAN-T5-Small | FLAN-T5-Base | FLAN-T5-XL | FLAN-T5-XXL |
|---|---|---|---|---|
| Cola-FT | 56.5 | 60.6 | 64.1 | 65.4 |
| Cola-Zero (2-shot) | 30.3 | 34.6 | 57.6 | 61.0 |
| Cola-Zero (0-shot) | 28.6 | 36.0 | 55.0 | 59.3 |

Table 7: **A-OKVQA validation set DA performance.** Extension of Figure 5.

| | 1-shot | 2-shot | 3-shot | 4-shot |
|---|---|---|---|---|
| Cola-Zero | 60.2 | 61.0 | 60.7 | 59.2 |

Table 8: **Cola-Zero in-context few-shot learning DA performance on A-OKVQA validation set.** Extension of Figure 6.

## A.5   Qualitative Examples

In this section, we provide more qualitative examples on A-OKVQA (Figure 8), e-SNLI-VE (Figure 9), and VSR (Figure 10) datasets.

Due to the large span of the three figures, for better visibility, we put the detailed description directly in each figure's caption part. We illustrate how Cola-FT and Cola-Zero  process the VLMs answers in each example. Overall, in these examples, we can observe that even if BLIP and OFA provide wrong answers, Cola can still present the correct answer based on the captions provided by OFA and BLIP, as well as the choice set. This may illustrate how Cola amazingly accomplishes visual reasoning tasks via coordinating BLIP and OFA.

## A.6   Failure Cases

In Figure 11, we provide a few failed cases to analyze the specific behavior of Cola.

The leftmost example's correct answer is *kayaking*, but there are no hints from OFA and BLIP's answers and captions. Therefore Cola-Zero incorrectly provides the answer *OFA* without sufficient information as hints, while surprisingly Cola-FT answered correctly from OFA's *boating* answer.

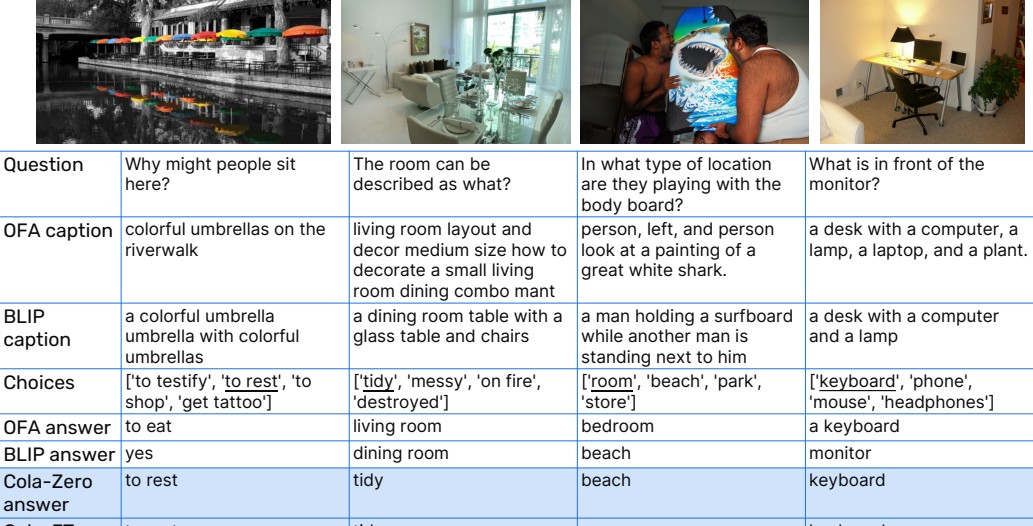

| | | | | |
|---|---|---|---|---|
| Question | Why might people sit here? | The room can be described as what? | In what type of location are they playing with the body board? | What is in front of the monitor? |
| OFA caption | colorful umbrellas on the riverwalk | living room layout and decor medium size how to decorate a small living room dining combo mant | person, left, and person look at a painting of a great white shark. | a desk with a computer, a lamp, a laptop, and a plant. |
| BLIP caption | a colorful umbrella umbrella with colorful umbrellas | a dining room table with a glass table and chairs | a man holding a surfboard while another man is standing next to him | a desk with a computer and a lamp |
| Choices | ['to testify', 'to rest', 'to shop', 'get tattoo'] | ['tidy', 'messy', 'on fire', 'destroyed'] | ['room', 'beach', 'park', 'store'] | ['keyboard', 'phone', 'mouse', 'headphones'] |
| OFA answer | to eat | living room | bedroom | a keyboard |
| BLIP answer | yes | dining room | beach | monitor |
| Cola-Zero answer | to rest | tidy | beach | keyboard |
| Cola-FT answer | to rest | tidy | room | keyboard |

Figure 8: **A-OKVQA qualitative examples.** Leftmost: LLM doesn't use BLIP and OFA's answers, but may observe from captions to derive the correct final answer. Left: As shown on the left, LLM does not follow the wrong answers from OFA and BLIP but gets the correct answers from captions. Right: With both OFA and BLIP answering incorrectly, LLM derives the correct one from both VLMs' captions and answers. Rightmost: After assessing the questions, answers, and captions, LLM goes with OFA's answer and rewrites it to match the expression in the choices. The correct choices are underlined. Cola-Zero answers are given in zero-shot settings.

The left example again has insufficient information from captions. While BLIP answers *no* and OFA answers *yes*, Cola-FT chooses to answer *maybe*, which looks natural but unfortunately picks the wrong choice.

The right example's captions contain enough information this time. But both Cola-FT and Cola-Zero are misled by BLIP's wrong answer *no parking*.

The rightmost example also has insufficient information from captions. In this situation, Cola has no choice but to believe either BLIP or OFA's answer, but it mistakenly prefers BLIP's wrong answer.

## A.7 Prompt Templates

Across three datasets, the prompt template is roughly the same, with minor differences mainly in the format of the questions and choices. We list the prompt templates adopted in A-OKVQA and e-SNLI-VE/VSR in Table 9 and Table 10, respectively.

## A.8 Parameter-efficient Finetuning

To further reduce the computation cost in model adaptation, we explored parameter-efficient finetuning (PEFT) techniques to reduce finetuning parameter counts. Specifically, we use $(IA)^3$ [57], which finetunes an overhead of 1 million parameters, equivalent to 0.01% of the full parameters of FLAN-T5-XXL.

Compared to full finetuning, $(IA)^3$ requires more iterations to converge. The performance of a $(IA)^3$ finetuned FLAN-T5-XXL model is on par with a fully finetuned FLAN-T5-Small (80 million parameters) counterpart (Figure 5). Notably, the former is associated with more computation and memory footprint as a consequence of more parameters in the forward pass.

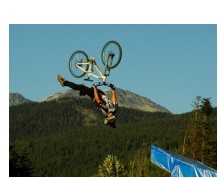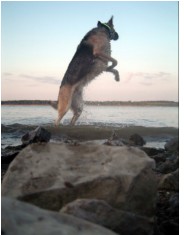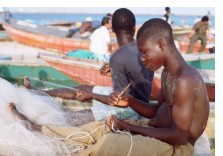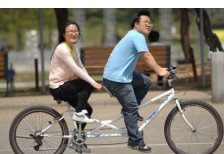

| | | | | |
|---|---|---|---|---|
| Question | Does the image describe " A professional daredevil "? | Does the image describe " the dog is a shitz " ? | Does this image describe "Two twenty-somethings prepare to catch salmon while other older men catch catfish" ? | Does this image describe "A little girl gets hit by a woman riding a bike" ? |
| OFA caption | person doing a flip on a mountain bike | a dog jumping out of the water. | men repairing fishing nets on the beach in zanzibar, tanzania | a man and a woman on a tandem bike |
| BLIP caption | a man doing a trick on a bike in the air | a dog jumping over rocks in the water | a man sitting on a boat with a fishing net net | a man and woman riding a bicycle in a parking lot |
| Choices | ['yes', 'maybe', 'no'] | ['yes', 'maybe', 'no'] | ['yes', 'maybe', 'no'] | ['yes', 'maybe', 'no'] |
| OFA answer | yes | no | yes | yes |
| BLIP answer | yes | no | yes | no |
| Cola-Zero answer | yes | no | no | no |
| Cola-FT answer | maybe | maybe | maybe | no |

Figure 9: **e-SNLI-VE qualitative examples.** Leftmost: As the connection to *daredevil* is not obvious in BLIP and OFA's captions, although Cola-Zero is misled, Cola-FT correctly answers *maybe*. Left: Similar to the left example, Cola-FT answer correctly as no obvious connections are seen from the captions to this question. Right: Similar to the left example, the fact of *catch catfish* is not reasonable from the captions, Cola-FT picks the correct answer *maybe*. Rightmost: As *girl gets hit* is not obvious in BLIP and OFA's captions and answers, Cola-Zero and Cola-FT both follow BLIP to choose the correct answer *no*. The correct choices are underlined. Cola-Zero answers are given in zero-shot settings.

## A.9 Extended Ablation Studies

**Do caption labels offer useful information to LLM? How would more prompt variations affect the performance of Cola?** We tested Cola-Zero with and without caption labels on A-OKVQA validation set, observing a slight decrease in performance when without them (70.39% w/t vs. 69.97% w/o). More ablative experiments showed that removing the VLM's answer labels led to a substantial drop in performance (70.39% w/t vs. 67.62% w/o). Removing the model characteristic descriptions also led to a decrease (70.39% w/t vs. 68.37% w/o).

**Do longer image captions improve reasoning performance?** On A-OKVQA validation set, we tested longer image descriptions (>50 tokens) but found no gain compared to Cola or single VLMs. Longer captions decreased FLAN-T5+OFA's accuracy by 0.61% and FLAN-T5 with BLIP by 0.69% on the A-OKVQA validation set. Cola (captions <30 tokens) reached 77.73%, outperforming individual VLMs. Longer captions lacked meaningful visual context, possibly due to short text and image pairs in their training datasets. This experiment reaffirms Cola's effectiveness in aggregating individual VLM functionalities.

## B Extended Related Works

### B.1 Finetuning Large Language Models

Large language models [8, 71, 6] pretrained on massive amounts of unstructured data have gradually demonstrated great performance by finetuning on additional task-specific instances. Finetuning a large language model can be considerably more sample efficient than re-training from scratch, although acceptable performance may still require a considerable quantity of data [95]. Recent

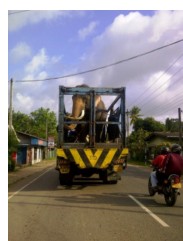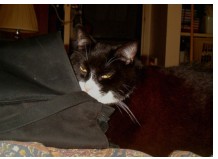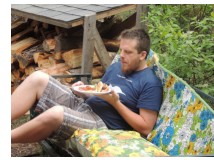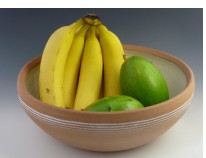

| | | | | |
|---|---|---|---|---|
| Question | Does this image describe "The truck contains the elephant" ? | Does this image describe "The bed is under the handbag" ? | Does this image describe "The couch is behind the hot dog" ? | Does this image describe "The bowl contains the banana" ? |
| OFA caption | an elephant being transported on a truck in sri lanka | a black and white tuxedo cat with a white nose, yellow eyes, and white | person enjoying a meal by the fire | bananas and mangoes in a bowl |
| BLIP caption | a truck with a large elephant in the back of it | a black cat laying on a bed with a pillow | a man sitting on a couch with a plate of food | a bowl of fruit is shown in this bowl |
| Choices | ['yes', 'no'] | ['yes', 'no'] | ['yes', 'no'] | ['yes', 'no'] |
| OFA answer | yes | no | yes | yes |
| BLIP answer | no | no | yes | no |
| Cola-Zero answer | no | no | no | yes |
| Cola-FT answer | yes | no | no | yes |

Figure 10: **VSR qualitative examples.** Leftmost: As OFA caption mentioned *elephant being transported* and OFA provides the correct answer, Cola-FT follows OFA's choice. Left: As OFA and BLIP provide the same answer, Cola-Zero and Cola-FT follow the choice. Right: As the captions do not provide obvious information, even BLIP and OFA provide the same answer, Cola-Zero and Cola-FT are not misled to the wrong choice. Rightmost: As the captions provide strong clue *bananas in a bowl*, although BLIP's answer is incorrect, Cola-Zero and Cola-FT still choose the correct answer. The correct choices are underlined. Cola-Zero answers are given in zero-shot settings.

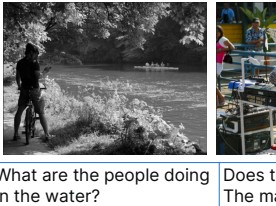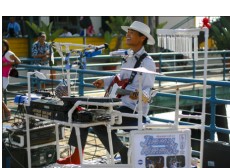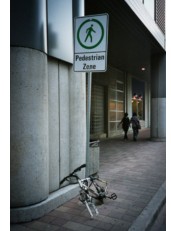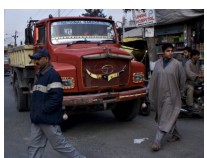

| | | | | |
|---|---|---|---|---|
| Question | What are the people doing in the water? | Does the image describe " The man is making a vase"? | What kind of zone is this bike parked in? | Does this image describe "The motorcycle is beside the truck" ? |
| OFA caption | black and white photo of a man on a bike looking at a canoe in the river | person on the potter's wheel | a city made by people bucharest | men walking past a truck in kabul, afghanistan. |
| BLIP caption | a man and woman on a bike in a park | a man is sitting on a chair and is using a wheel | a bicycle parked next to a pedestrian crossing sign | a man walking down the street in a city |
| Choices | ['surfing', 'fishing', 'kayaking', 'swimming'] | ['yes', 'maybe', 'no'] | ['temporary', 'pedestrian', 'no parking', 'handicap'] | ['yes', 'no'] |
| OFA answer | boating | yes | pedestrian | yes |
| BLIP answer | swimming | no | no parking | no |
| Cola-Zero answer | OFA | no | no parking | no |
| Cola-FT answer | kayaking | maybe | no parking | no |

Figure 11: **Failed cases.** The correct choices are underlined. Cola-Zero answers are given in zero-shot settings.

works have finetuned task-specific models that demonstrate amazing capabilities in many real-world applications, such as Copilot for program synthesis [11].

---

**VQA Prompt Template**

---

Answer the following multiple-choice question by OFA and BLIP's description and their answers to the visual question. OFA and BLIP are two different vision-language models to provide clues.

OFA's description: `<OFA caption>`
BLIP's description: `<BLIP caption>`

Q: `<Question>`

OFA's answer: `<OFA answer>`
BLIP's answer: `<BLIP answer>`

Choices: `<Choices to the question>`

A:

---

Table 9: **VQA prompt template for the LLM, for VQA v2 / OK-VQA / A-OKVQA.** The LLM is instructed to coordinate VLMs. Each question set defines *visual context*, *question with choices*, and *plausible answers*.

---

**e-SNLI-VE / VSR Prompt Template**

---

Answer the following multiple-choice question by OFA and BLIP's description and their answers to the visual question. OFA and BLIP are two different vision-language models to provide clues.

OFA's description: `<OFA caption>`
BLIP's description: `<BLIP caption>`

Q: does the image describe `<hypothesis>` ?

OFA's answer: `<OFA answer>`
BLIP's answer: `<BLIP answer>`

e-SNLI-VE Choices: [yes, no, maybe]
VSR Choices: [yes, no]

A:

---

Table 10: **e-SNLI-VE/VSR prompt template for the LLM.** The LLM is instructed to coordinate VLMs. Each question set defines *visual context*, *hypothesis*, and *plausible answers*.

## B.2 Instruction-based Learning

Recent advances in the capabilities of language models have piqued researchers' curiosity in the field of instruction-based learning [26, 64, 87, 24]. The core of instruction-based learning is to explore the knowledge of the language model itself. In contrast to prompt learning to stimulate the language model's ability to complete blanks, instruction tuning more focuses on activating the language model's comprehension by giving obvious instructions to models and expecting correct feedback. Earlier work [67] finetune BART [46] using instructions and few-shot exemplars in question answering, text classification, and text modification. Their findings suggest that few-shot instruction tuning improves performance on unseen tasks. [66] finetunes GPT-2 Large and also observes that few-shot exemplar instruction tuning could improve performance. [85] finetunes T5-11B with more diverse instruction templates and observe similar improvements in zero-shot learning. More recent work [108] performs large-scale experiments with a 137B FLAN-T5 model and instruction-tune it on over 60 datasets verbalized via instruction templates. They observe FLAN-T5 substantially improves over zero-shot GPT-3 (175B) on 20 of 25 evaluation datasets. OpenAI also releases InstructGPT [71] based on GPT-3 [8], it makes use of human annotations to steer desired model behavior through both instruction

|  | Accuracy | # Finetuning Params |
|---|---|---|
| Finetuning | 77.73 | 11B (100%) |
| PEFT, $(\text{IA})^3$ | 63.76 | 1M (0.01%) |

Table 11: $(\textbf{IA})^3$ [57] **parameter-efficient tuning (PEFT) performance.** We finetune a FLAN-T5-XXL model on the A-OKVQA training set and evaluate it on the A-OKVQA validation set.

tuning and reinforcement learning of human feedback. They discover that InstructGPT is favored by humans over unmodified GPT-3.

### B.3 Visual Reasoning

Beyond the uni-modal reasoning tasks such as question answering (QA) [100, 41, 14, 80, 79, 23, 75, 17, 96, 25, 127, 119, 7], visual reasoning requires models to not only understand and interpret visual information but also to apply high-level cognition to derive rational solutions [40, 35, 4, 59, 60, 84, 120, 34]. Several tasks have been introduced to address visual reasoning, such as visual question answering (VQA) [1], in which models are expected to provide answers to questions related to an image and visual entailment (VE) [113], where the model is required to determine the similarity or relationship between a given image and a description. Classic visual reasoning methods have employed an image encoder and a text encoder, along with a reasoning block that utilizes attention mechanisms [121, 72, 122, 105], neuro-symbolic methods [117, 61, 116], or external knowledge [62, 28, 13] to perform reasoning.

Recent progress in large pre-trained models has led to the development of language models (LLMs) that possess exceptional commonsense reasoning capabilities [78, 16, 15, 77]. These models can potentially replace the reasoning block in visual reasoning tasks, and LLMs' lack of perception can be compensated by incorporating multiple vision-language models (VLMs) trained on different domains [76, 104, 52]. For example, PICa [115] converts the image into captions that GPT-3 [8] can understand, and adapts GPT-3 to solve the VQA task in a few-shot manner by providing a few in-context VQA examples. However, there is still a lack of research on how to harness the collective power of these complementary VLMs for visual reasoning tasks.

### B.4 Model Ensembling

Model ensembling is a powerful machine learning technique that combines the predictions of multiple models to improve the overall performance of a given task [20]. Classic model ensembling methods include simple averaging, weighting the predictions based on model performance, and stacking the models. By combining the predictions of multiple models, ensembling can reduce the variance and bias of the final predictions, resulting in a more robust and accurate model [83]. Ensemble methods have been shown to perform well in a wide range of tasks, including image classification, natural language processing, and time series forecasting. However, when it turns to multimodal tasks such as visual reasoning, a simple combination is not applicable to heterogeneous models as their inputs and outputs vary.

The Mixture-of-Experts (MoE) [91, 82, 126, 45, 47] can be conceptualized as a model ensemble strategy implemented at the level of network architecture. MoE-based multi-modal models [33] excel in leveraging the specific strengths of each expert, thereby delivering the performance that often outstrips that of any individual expert. In these networks, the credibility of each expert's output is dynamically weighted, facilitating a comprehensive and nuanced response to multimodal tasks.

However, even within this sophisticated framework, challenges can arise, particularly when managing heterogeneous pre-trained multimodal models. To address this problem, an innovative approach known as Socratic Models (SMs) [123] has been proposed. SMs employ prompt engineering to guide these diverse models through multimodal discussions, effectively combining their varied knowledge. This method promotes a more harmonious and effective integration of different models, enhancing the ensemble's ability to handle complex tasks.

With a similar goal, [54] proposes a closed-loop iterative consensus optimization method to utilize the strengths of individual models. However, previous methods do not fully explore the potential of a centralized solution or adapt to the separate functionalities of different models, particularly in the

visual reasoning scenario. Recent studies, such as CICERO [65], have shown that language models possess strong capabilities in coordinating multiple agents, which inspires us to reorganize pre-trained multimodal models with a focus on the language models.

## Broader Impact

This study inherits ethical risks of biases from pretrained VLMs and LLMs, depending on their training data. We suggest the users consider the possible biases in reasoning and prompt the model to interpret its predictions in natural languages when necessary.

