[33] Woojeong Jin, Yu Cheng, Yelong Shen, Weizhu Chen, and Xiang Ren. A good prompt is worth millions of parameters: Low-resource prompt-based learning for vision-language models. In *Proceedings of the 60th Annual Meeting of the Association for Computational Linguistics (Volume 1: Long Papers)*, pages 2763–2775, Dublin, Ireland, May 2022. Association for Computational Linguistics. 4

[34] Justin Johnson, Bharath Hariharan, Laurens Van Der Maaten, Li Fei-Fei, C Lawrence Zitnick, and Ross Girshick. Clevr: A diagnostic dataset for compositional language and elementary visual reasoning. In *Proceedings of the IEEE conference on computer vision and pattern recognition*, pages 2901–2910, 2017. 2, 9, 23

[35] Mandar Joshi, Eunsol Choi, Daniel S. Weld, and Luke Zettlemoyer. Triviaqa: A large scale distantly supervised challenge dataset for reading comprehension. In *ACL*, 2017. 23

[36] Maxime Kayser, Oana-Maria Camburu, Leonard Salewski, Cornelius Emde, Virginie Do, Zeynep Akata, and Thomas Lukasiewicz. e-vil: A dataset and benchmark for natural language explanations in vision-language tasks. In *Proceedings of the IEEE/CVF International Conference on Computer Vision*, pages 1244–1254, 2021. 4

[37] Daniel Khashabi, Yeganeh Kordi, and Hannaneh Hajishirzi. Unifiedqa-v2: Stronger generalization via broader cross-format training. *arXiv preprint arXiv:2202.12359*, 2022. 4

[38] Wonjae Kim, Bokyung Son, and Ildoo Kim. Vilt: Vision-and-language transformer without convolution or region supervision. In *International Conference on Machine Learning*, pages 5583–5594. PMLR, 2021. 4

[39] Mike Lewis, Shruti Bhosale, Tim Dettmers, Naman Goyal, and Luke Zettlemoyer. Base layers: Simplifying training of large, sparse models. In *International Conference on Machine Learning*, pages 6265–6274. PMLR, 2021. 24

[40] Mike Lewis, Yinhan Liu, Naman Goyal, Marjan Ghazvininejad, Abdelrahman Mohamed, Omer Levy, Ves Stoyanov, and Luke Zettlemoyer. Bart: Denoising sequence-to-sequence pre-training for natural language generation, translation, and comprehension. *arXiv preprint arXiv:1910.13461*, 2019. 23

[41] Bo Li, Yifei Shen, Jingkang Yang, Yezhen Wang, Jiawei Ren, Tong Che, Jun Zhang, and Ziwei Liu. Sparse mixture-of-experts are domain generalizable learners. *arXiv preprint arXiv:2206.04046*, 2022. 24

[42] Bo Li, Yuanhan Zhang, Liangyu Chen, Jinghao Wang, Jingkang Yang, and Ziwei Liu. Otter: A multi-modal model with in-context instruction tuning, 2023. 2

[43] Junnan Li, Dongxu Li, Silvio Savarese, and Steven Hoi. Blip-2: Bootstrapping language-image pre-training with frozen image encoders and large language models. *arXiv preprint arXiv:2301.12597*, 2023. 2, 4

[44] Junnan Li, Dongxu Li, Caiming Xiong, and Steven Hoi. Blip: Bootstrapping language-image pre-training for unified vision-language understanding and generation. *arXiv preprint arXiv:2201.12086*, 2022. 2, 3, 9, 18, 24

[45] Liunian Harold Li, Mark Yatskar, Da Yin, Cho-Jui Hsieh, and Kai-Wei Chang. Visualbert: A simple and performant baseline for vision and language. *arXiv preprint arXiv:1908.03557*, 2019. 4

[46] Shuang Li, Yilun Du, Joshua B. Tenenbaum, Antonio Torralba, and Igor Mordatch. Composing ensembles of pre-trained models via iterative consensus. *ArXiv*, abs/2210.11522, 2022. 9, 24

[47] Tsung-Yi Lin, Michael Maire, Serge Belongie, James Hays, Pietro Perona, Deva Ramanan, Piotr Dollár, and C Lawrence Zitnick. Microsoft coco: Common objects in context. In *European conference on computer vision*, pages 740–755. Springer, 2014. 18

[48] Fangyu Liu, Guy Emerson, and Nigel Collier. Visual spatial reasoning. *arXiv preprint arXiv:2205.00363*, 2022. 2, 18

[49] Haokun Liu, Derek Tam, Mohammed Muqeeth, Jay Mohta, Tenghao Huang, Mohit Bansal, and Colin Raffel. Few-shot parameter-efficient fine-tuning is better and cheaper than in-context learning. *arXiv preprint arXiv:2205.05638*, 2022. 20, 23

[50] Jiasen Lu, Dhruv Batra, Devi Parikh, and Stefan Lee. Vilbert: Pretraining task-agnostic visiolinguistic representations for vision-and-language tasks. *Advances in neural information processing systems*, 32, 2019. 4

[51] Jiasen Lu, Christopher Clark, Rowan Zellers, Roozbeh Mottaghi, and Aniruddha Kembhavi. Unified-io: A unified model for vision, language, and multi-modal tasks. *arXiv preprint arXiv:2206.08916*, 2022. 4