# OpenReview forum: "Large Language Models are Visual Reasoning Coordinators"
_NeurIPS.cc/2023/Conference — NeurIPS 2023 poster_

### Official Review · Reviewer_Yu6H · 2023-07-04

**Soundness:** 3 good
**Presentation:** 3 good
**Contribution:** 3 good
**Rating:** 7
**Confidence:** 4

**Summary:**

1. The author proposes to utilize a language model as the coordinator between different outputs from different VLMs, leveraging their strengths for visual reasoning.
2. The proposed method achieves SOTA results on multiple visual reasoning benchmarks.
3. The analysis shows how zero-shot / fine-tuned language models can coordinate VLM outputs.

**Strengths:**

1. The proposed method with zero-shot VLMs and zero-shot / fine-tuned LLMs as coordinators are novel.
2. The performance of the model is significant with sufficient experiment results.

**Weaknesses:**

1. Different dataset requires different visual / reasoning capabilities and knowledge. The proposed method performs differently on different datasets. For example, the ZS version performs similarly with 'Ensemble' on VQA v2 and OK-VQA, but outperforms 'Ensemble' significantly on other datasets. It's better to analyze the reason.
2. Why the proposed method performs better than other methods needs more discussion. For example, comparing with BLIP-2 on VQA.


**Questions:**

1. It is a little unclear that in Table 3, the Cola-FT model is fine-tuned on BLIP+OFA outputs or 3 OFA variance. Why is the Cola-FT performance here much lower than the performance in Table 2 and Figure 3?

**Limitations:**

The limitations are discussed and potential solutions are proposed.

---

> ### Author Rebuttal · Authors · 2023-08-08
>
> We kindly request the reviewer to check our detailed responses and revisions in the following. Your time, effort, and affirmation of our research are truly valued.
>
> **Q: Different dataset requires different visual / reasoning capabilities and knowledge….**
>
> We appreciate the reviewer's insightful question. This is inherently determined by the mechanism of Cola-Zero. It leverages few-shot in-context examples to guide the LLM in harmonizing VLM outputs to obtain the final accurate answer. Its efficiency largely hinges on the quality of these examples.
>
> Key factors to consider include:
>
> 1. **Question Format:** Datasets like VQA v2 and OK-VQA contain open-ended questions, while A-OKVQA, e-SNLI-VE, and VSR use multiple-choice. Converting VQA v2 and OK-VQA to classification introduces complexities for traditional ensemble methods, as evident in Table 2. Classic methods struggle with generative models like API-based GPT-4, underscoring Cola's value as an end-to-end ensemble strategy for extensive (vision-)language models. Moreover, Cola-Zero’s efficiency also relies on the question format – it's easier for LLMs to answer when given choices like in A-OKVQA. Conversely, Cola-FT finetunes LLMs to discern answer formats.
> 2. **Knowledge Demands and Out-of-Distribution Challenges:** Datasets requiring deep reasoning, such as A-OKVQA, emphasize Cola-Zero and FT’s advantages over Ensemble. With A-OKVQA demanding broad commonsense and world knowledge, pretrained LLMs (like our FLAN-T5) are favored. On such datasets, Cola-Zero outperforms the ensemble baseline, while established models like OFA and BLIP-2, trained on VQA v2, set higher baselines, making significant improvements by Cola challenging.
> 3. **In-context Examples:** Cola-Zero's performance is influenced by its in-context examples. Misrepresentative examples can hamper its ability to synchronize VLMs effectively. We use random selection as our baseline for this learning. However, Cola-FT overcomes such limitations. By finetuning, it offers a comprehensive grasp on harmonizing VLM outputs, consistently delivering excellent results, and potentially matching top-tier results.
>
> **Q: Why the proposed method performs better than other methods needs more discussion…**
>
> Compared to individual models like BLIP and OFA, Cola's strength is its integration of responses from multiple models, refined by a language model, to produce superior results. For instance, Figure 2's third example shows that while OFA and BLIP individually fail, Cola-FT uses their captions to deduce the correct answer, “maybe”. While Cola's efficacy may vary across datasets, its LLM can harness comprehensive knowledge to improve VLM outputs. For example, the left side of Figure 8 in the supplementary material showcases instances where both VLMs err. Yet, both Cola-Zero and Cola-FT interpret the data to deliver the right answer, “to rest”. While Cola-Zero's effectiveness is limited by dataset question formats and context examples, Cola-FT's fine-tuning broadens its adaptability across diverse questions. See also our response to Reviewer 3ict.
>
> **Q: It is a little unclear that in Table 3, the Cola-FT model is fine-tuned on BLIP+OFA outputs or 3 OFA variance…**
>
> We greatly appreciate this constructive question raised by the reviewer, which warrants further clarification. In Table 3, we run OFA-base models under different random seeds, represented as OFA-Base-1, OFA-Base-2, etc. Therefore, the corresponding results of Ensemble and Cola-Zero/FT in Table 3 are based on the outputs of the aforementioned OFA-Base-1/2/3 models. Similarly, in Table 4, the structure of Ensemble and Cola-Zero/FT is based on the outputs of the OFA-tiny/medium/base models.
>
> In both experiments, the BLIP model is not involved and the prompt is slightly changed to accommodate more VLMs. We have clarified the settings in our revised paper.
>
> As an ensemble method, Cola demonstrates larger performance gains compared to single VLMs when the VLMs are more complementary in their capabilities. Table 3 presents an example where the three VLMs are nearly identical. As a consequence, the performance gains of Cola-Zero and Cola-FT are limited, and so are Ensemble baselines. In Table 4, OFA-tiny/medium/base possesses different capabilities, which results in a much larger performance gain for Cola-Zero (3.6% on A-OKVQA, 5.0% on e-SNLI-VE) and Cola-FT (8.5% on A-OKVQA, 11.1% on e-SNLI-VE). In Table 2, BLIP and OFA models are quite different in their pre-training data so they show complementary capabilities (see examples in Figure 2, 8-11). Therefore, the performance gains for Cola-Zero and Cola-FT are even larger.

---

> > ### Author Response · Authors · 2023-08-14
> > **Sincerely Looking Forward to Your Reply**
> >
> > Dear Reviewer,
> >
> > We extend our heartfelt gratitude for the invaluable suggestions and comments you have provided, which have significantly contributed to refining our paper. Specifically, your insights on the performance gap between Cola-Zero and ensemble baselines helped us better present our analysis of why Cola-Zero works. We hope our response has addressed your concerns.
> >
> > We eagerly anticipate your response to our revised submission and are earnestly open to engaging in any discourse aimed at enhancing the quality of our paper.
> >
> > Warm regards,
> >
> > The authors

---

### Official Review · Reviewer_qcEV · 2023-07-07

**Soundness:** 3 good
**Presentation:** 3 good
**Contribution:** 3 good
**Rating:** 7
**Confidence:** 4

**Summary:**

The paper proposes an ensemble based approach to solve visual reasoning problems. The paper proposes to use an instruction fine-tuned large language model to integrate answers to visual reasoning problems provided by vision language models. The paper presents two variants of the aggregation model -- using fine-tuning and using in-context learning. The model is evaluated on the VQA v2, A-OKVQA, OK-VQA, e-SNLI-VE and VSR datasets.

**Strengths:**

* The paper reports promising results on the VQA v2, A-OKVQA, OK-VQA, e-SNLI-VE and VSR datasets.
* The paper includes a variety of ablations that show the effectiveness of the proposed method -- including model size, scaling, number of video-language models as ensemble members.
* The paper includes qualitative examples which highlight the effectiveness of the proposed method.
* The paper is well written and easy to understand.

**Weaknesses:**

* While the results are very promising, it would be helpful to add results on more complex datasets such as CLEVR (CLEVR: A Diagnostic Dataset for Compositional Language and Elementary Visual Reasoning) and GQA (GQA: A New Dataset for Real-World Visual Reasoning and Compositional Question Answering) which study compositional reasoning.

* The paper should also consider comparing to prior work such as Visual Programming: Compositional visual reasoning without training, CVPR 2023, which also uses an external large language model to coordinate vision / language-vision models.

* The paper claims in L291 "This work demonstrates the first step toward applying language models for visual reasoning", but Flamingo (Flamingo: a Visual Language Model for Few-Shot Learning, NeurIPS 2022) already shows zero-show visual reasoning on datasets such as Next-QA.



**Questions:**

* The paper should include further details on the computational resources used. L680 in the supplementary material just states that V100 or A100 GPUs were used, but the paper should include further details about the total computational resources used.

* The paper should include further motivational details on why the particular datasets VQA v2, A-OKVQA, OK-VQA, e-SNLI-VE and VSR were used? Why were more datasets which require more complex reasoning abilities such as CLEVR and GQA were not used.

**Limitations:**

The paper does not include any discussion about its limitations.

---

> ### Author Rebuttal · Authors · 2023-08-08
>
> We appreciate the reviewer for your feedback on our paper, and we will provide our responses to these concerns in the following. We have also made targeted modifications to the paper, and these issues will help improve our paper.
>
> Our work aims to demonstrate that by using LLM, multiple VLMs can be coordinated to achieve better visual reasoning effects. We hope this work can bring some of our insights to the currently booming fields of LLM and VLM. We hope the reviewer will carefully review our responses and modifications, and we sincerely thank the reviewer for your time and effort, as well as your acknowledgement of our work.
>
> **Q: CLEVR and GQA results (The paper should include further motivational details on why the particular datasets VQA v2, A-OKVQA, OK-VQA, eSNLI-VE and VSR were used? Why were more datasets which require more complex reasoning abilities such as CLEVR and GQA were not used.)**
>
> **Q: The paper should also consider comparing to prior work such as Visual Programming: Compositional visual reasoning without training, CVPR 2023, which also uses an external large language model to coordinate vision / language-vision models.**
>
> We thank the reviewer for the suggestion of experiments on compositional reasoning datasets and Visual Programming. Below we release the results on GQA and CLEVR datasets. On the GQA validation set, Cola-FT shows marginal improvement over best VLM (OFA), and outperforms Visual Programming [1].
>
> | GQA | Acc. | $\Delta$ |
> | --- | --- | --- |
> | BLIP | 41.7 |  |
> | OFA | 58.0 |  |
> | Cola-FT | 60.3 | +2.32 |
> | VisProg [1] | 50.5 |  |
>
> On the CLEVR validation set, Cola-Zero shows marginal performance gain and Cola-FT shows substantial performance gain over single VLMs. It’s interesting to note that InstructBLIP (FLAN-T5-XL) [2] outperforms InstructBLIP (FLAN-T5-XXL) by a large margin in the 0-shot evaluation.
>
> | CLEVR | Acc. | $\Delta$ |
> | --- | --- | --- |
> | InstructBLIP (FLAN-T5-XL) | 33.7 |  |
> | InstructBLIP (FLAN-T5-XXL) | 16.6 |  |
> | Cola-Zero (2-shot) | 34.4 | +0.7 |
> | Cola-FT | 54.3 | +20.6 |
>
> [1] Gupta, T., & Kembhavi, A. (2023). Visual programming: Compositional visual reasoning without training. In *Proceedings of the IEEE/CVF Conference on Computer Vision and Pattern Recognition* (pp. 14953-14962).
>
> [2] Dai, W., Li, J., Li, D., Tiong, A.M., Zhao, J., Wang, W., Li, B., Fung, P., & Hoi, S. (2023). InstructBLIP: Towards General-purpose Vision-Language Models with Instruction Tuning. *ArXiv, abs/2305.06500*.
>
> **Q: The paper claims in L291 "This work demonstrates the first step toward applying language models for visual reasoning", but Flamingo (Flamingo: a Visual Language Model for Few-Shot Learning, NeurIPS 2022) already shows zero-show visual reasoning on datasets such as Next-QA.**
>
> We greatly appreciate the reviewer pointing out the oversight regarding the 'Flamingo' work in our article. Our claim, 'This work demonstrates the first step toward applying language models for visual reasoning,' is primarily based on our unique approach, where we employ a standalone and end-to-end LLM  to coordinate outputs from various VLMs and combine them for enhanced results. This, compared to either a single VLM or multiple VLM ensembles, led to significant improvements in multiple visual reasoning datasets as observed with our 'Cola-Zero/FT'. Thus, a better way to quote our claim would be "This work demonstrates the first step toward applying end-to-end language models for visual reasoning", which we have revised in the paper.
>
> On this front, compared to VLM models that incorporate powerful LLMs—such as Flamingo which can integrate with OPT[1] or Chinchilla [2], and OpenFlamingo [3] which can combine with LLaMA [4]—our Cola strategy could potentially benefit from merging with a larger scale LLM to interpret VLM outputs, leading to even better outcomes. For example, Cola-FT obviously applies to end-to-end LM APIs (like Open GPT-4 [5] and Anthropic Claude [6]), which is challenging for conventional ensemble methods or VLMs that need to be finetuned from LLMs.
>
> [1] Zhang, S., Roller, S., Goyal, N., Artetxe, M., Chen, M., Chen, S., ... & Zettlemoyer, L. (2022). Opt: Open pre-trained transformer language models. *arXiv preprint arXiv:2205.01068*.
>
> [2] Hoffmann, J., Borgeaud, S., Mensch, A., Buchatskaya, E., Cai, T., Rutherford, E., ... & Sifre, L. (2022). Training compute-optimal large language models. *arXiv preprint arXiv:2203.15556*.
>
> [3] Awadalla, A., Gao, I., Gardner, J., Hessel, J., Hanafy, Y., Zhu, W., ... & Schmidt, L. (2023). OpenFlamingo: An Open-Source Framework for Training Large Autoregressive Vision-Language Models. *arXiv preprint arXiv:2308.01390*.
>
> [4] Touvron, H., Lavril, T., Izacard, G., Martinet, X., Lachaux, M. A., Lacroix, T., ... & Lample, G. (2023). Llama: Open and efficient foundation language models. *arXiv preprint arXiv:2302.13971*.
>
> [5] OpenAI (2023). GPT-4 Technical Report. *ArXiv, abs/2303.08774*.
>
> [6] Anthropic \ Introducing Claude https://www.anthropic.com/index/introducing-claude
>
> **Q: the paper should include further details about the total computational resources used.**
>
> We greatly appreciate the reviewer for pointing out the need for additional detail in this section. To address this, we've incorporated a table illustrating the GPU hours as the computation cost in our training process. These computations are performed with reference to a server with 8 NVIDIA V100 GPUs.
>
> | Cola-FT | V100 hours |
> | --- | --- |
> | A-OKVQA | 12 |
> | e-SNLI-VE | 8 |
> | VSR | 8 |
> | GQA | 24 |
> | VQA v2 | 80 |
> | OK-VQA | 12 |
> | CLEVR | 24 |
>
> For most datasets, the inference time of Cola-Zero and Cola-FT is ~16 questions / second, with 1 A100 GPU. Each question is composed of 90 to 150 tokens. We have included these details in our revised paper.

---

> > ### Comment · Reviewer_qcEV · 2023-08-11
> > **Update**
> >
> > The results on CLEVR and GQA are promising. The final version should discuss prior work in more detail and update claims accordingly. I would keep my score and vote for acceptance.

---

> > > ### Author Response · Authors · 2023-08-12
> > > **Genuinely Thankful for Reviewers Feedback**
> > >
> > > Thanks! your reviews and suggestions significantly help us to improve our work. We hope our effort can bring value to the research community.

---

### Official Review · Reviewer_3ict · 2023-07-07

**Soundness:** 3 good
**Presentation:** 3 good
**Contribution:** 2 fair
**Rating:** 5
**Confidence:** 4

**Summary:**

The paper introduces a new paradigm called Cola, which aims to coordinate multiple vision-language models (VLMs) for visual reasoning tasks. While several VLMs have demonstrated strong commonsense reasoning abilities in different domains, effectively combining their capabilities remains a challenge. Traditional methods like ensembling struggle to achieve higher-order communications between these models.

**Strengths:**

Cola proposes a solution by employing a language model (LM) to coordinate the multiple VLMs. The LM facilitates natural language communication, leveraging the distinct and complementary capabilities of each VLM. The authors introduce two variants of Cola: Cola-FT, which involves fine-tuning the models, and Cola-Zero, which performs in-context learning without the need for fine-tuning.

The authors conduct extensive experiments to evaluate the performance of Cola on various visual reasoning tasks, including visual question answering (VQA), outside knowledge VQA, visual entailment, and visual-spatial reasoning. They demonstrate that Cola-FT achieves state-of-the-art results in these tasks. Additionally, Cola-Zero exhibits competitive performance in zero and few-shot settings, without requiring fine-tuning. The paper further includes ablation studies and visualizations to validate the effectiveness of the coordinator LM. These analyses confirm that the coordinator LM comprehends the instruction prompts and understands the individual functionalities of the VLMs, allowing it to coordinate their efforts and enable visual reasoning capabilities.

Using a language model as a coordinator for different VLMs is novel.

Adequate experiments show a promising performance over baselines.

**Weaknesses:**

Does the language coordinator play a role to determine the correctness of comparing multiple VLMs according to the answering language description? What if both VLMs are wrong?

As is mentioned in the limitation, no rational explanation or logical steps were applied either in
VLMs or LM.

The paper missed the recent work in visual reasoning, like Flamingo(https://nips.cc/Conferences/2022/ScheduleMultitrack?event=54165), STAR (http://star.csail.mit.edu/), GAMR(https://openreview.net/pdf?id=iLMgk2IGNyv), SHG-VQA(https://arxiv.org/abs/2304.08682), etc.

Only two VLMs are applied. It’s interesting to see the margin gains of Cola when compared with the best VLMs in the experiments

**Questions:**

Refer to the above comments

**Limitations:**

Refer to the above comments

---

> ### Author Rebuttal · Authors · 2023-08-08
>
> We would like to extend our gratitude to reviewer’s insightful critique of our paper. These thoughtful comments have guided revisions that will undoubtedly enhance the quality of our work. We hope that by highlighting the potential of language models (LMs) to coordinate multiple vision-language models (VLMs), our findings can provide new insights into large pretrained models and visual reasoning.
>
> **Q: Does the language coordinator play a role to determine the correctness of comparing multiple VLMs according to the answering language description? What if both VLMs are wrong?**
>
> **Q: As is mentioned in the limitation, no rational explanation or logical steps were applied either in VLMs or LM.**
>
> Yes, the language coordinator determines the correctness of VLM plausible answers implicitly, given their captions and the caption and answer labels (see Figure 3 and Sec 3.4). It’s all plausible answers, captions, VLM answer/caption labels, and the world knowledge the LM it encodes in itself, that contribute to the final decision of the language coordinator. Even in the cases where both VLMs give wrong answers, there are chances the language model arrives at the right answer.
>
> In the paper, we did not apply techniques such as Generating Rationale or Chain-of-Thought to make the LM’s logical process more explicit. However, we attempted in the supplementary experiments, on a small scale, to modify the prompt template of Cola-FT so that the model would output the logical conduction process, allowing us to observe the specific behaviors of the LM during its coordination of VLMs. We finetuning Cola-FT to output rationales before answers, by A-OKVQA ground truth rationales. In the modified prompt, we ask the model to provide rationales.
>
> Cola FT's demo output is as follows:
>
> - Case 1: leftmost example of Figure 8
>
> > Rationale: People might sit here to rest. The umbrellas are on the riverwalk. The answer is:rest
>
> OFA gives a reasonable answer (but out of choices) to the question while BLIP gives an irrelevant answer. In this case, both answers are wrong. However, either Cola-Zero or Cola-FT is able to infer from captions and plausible answers to give the correct answer “to rest”. The rationale suggests that the LM understands the scene that the umbrellas are on the riverwalk and guesses that people might sit here to rest based on commonsense. The final answer is correct.
>
> - Case 2: leftmost example of Figure 11
>
> > Rationale: The people are in a canoe. The people are in a canoe. The people are in a canoe. The answer is:kayaking
>
> Both VLMs are wrong in their plausible answers. OFA’s answer “boating” is semantically correct as the correct answer is “kayaking”, though it’s not the correct answer because this is a multiple-choice question. Cola-Zero gives a wrong answer “OFA” which is obviously wrong because “OFA” is the name of one of the VLMs given in the prompt and it’s out of the choices too. However, Cola-FT gives the correct answer “kayaking”, recognizing the correct choice based on prompts of captions and plausible answers after being finetuned. Even though the OFA and BLIP captions fail to identify that the people in the water are on a canoe. The LM identifies that the people in the water are associated with the canoe. The rationale is valid and helpful, though repetitive. The final answer is correct.
>
> - Case 3: third example of Figure 11
>
> > Rationale: The bike is parked in a no parking zone. The bike is parked next to a pedestrian crossing sign. The answer is:no parking
>
> From the rationale, we can tell that the LM understands that the bike is parked next to a pedestrian crossing sign. However, it “overthinks” that this is a no parking zone and therefore gives the wrong answer. This rationale helps us understand why Cola-FT gives a wrong answer.
>
> The inference results on A-OKVQA validation set are as follows:
>
> |  | Acc. |
> | --- | --- |
> | w/t rationale | 74.3 |
> | w/o rationale | 77.7 |
>
> To force the LM to output rationale does not improve the reasoning performance of Cola. This might be attributed to the low-quality ground truth rationales provided by the A-OKVQA dataset that we use to train the LM. Such rationales are just short and objective descriptions of the scene, without suggesting the underlying outside knowledge to answer the question. Therefore, training the LM to output rationale is harmful, though it outputs insights into the LM's behaviors during reasoning.
>
> **Q: The paper missed the recent work in visual reasoning, like Flamingo, STAR, GAMR, SHG-VQA, etc.**
>
> We appreciate the reviewer for pointing out this issue. We believe the related works you mentioned here are indeed relevant papers in the field of visual reasoning. Even though we have encompassed a citation list of total 120+ papers in our revision, it is hard to cover all relevant works in this rapidly and vigorously developing area. We have incorporated the related works mentioned above. This ensures that our audience can better reference more excellent papers by reading our paper.
>
> **Q: Only two VLMs are applied. It’s interesting to see the margin gains of Cola when compared with the best VLMs in the experiments.**
>
> We benchmark the best VLMs on A-OKVQA, InstructBLIP (FLAN-T5-XXL), and InstructBLIP (FLAN-T5-XL). For InstructBLIP baselines, we use InstructBLIP solely (i.e., without LM) to evaluate 0-shot performance. Interestingly, InstructBLIP (FLAN-T5-XL) slightly outperforms InstructBLIP (FLAN-T5-XXL). Referring to the results below, both Cola-Zero and Cola-FT improved the reasoning performance by substantial margins compared to single VLMs.
>
> |  | Acc. | $\Delta$ |
> | --- | --- | --- |
> | InstructBLIP (FLAN-T5-XL) | 60.4 |  |
> | InstructBLIP (FLAN-T5-XXL) | 59.8 |  |
> | Cola-Zero (0-shot) | 68.0 | +7.6 |
> | Cola-Zero (2-shot) | 72.3 | +11.9 |
> | Cola-FT | 78.1 | +17.7 |

---

> > ### Author Response · Authors · 2023-08-14
> > **Sincerely Looking Forward to Your Reply**
> >
> > Dear Reviewer,
> >
> > We sincerely appreciate you taking the time to provide thoughtful suggestions and comments, which have been immensely helpful in improving our paper. In particular, your feedback regarding the coordinator LM and Cola with the best VLMs has enabled us to strengthen and clarify these sections. We have revised the writing and added experiment results to our paper.
> >
> > We sincerely look forward to hearing your perspective on our response. Please know that we remain open to any discussion that could further enhance our work, and we highly value your constructive input.
> >
> > Sincerely,
> >
> > The authors

---

> > ### Comment · Reviewer_3ict · 2023-08-18
> > **Thank you for your responses**
> >
> > Hi Thank you for your responses. Although I have a different opinion on the first question, some of the questions were resolved by the responses.
> > But there is no revised version of the paper on the system, please check if it is uploaded successfully.
> > Will add scores to reflect the change.

---

> > > ### Author Response · Authors · 2023-08-18
> > > **Thank You for Recognizing Our Rebuttal**
> > >
> > > Dear Reviewer,
> > >
> > > Thank you for your reply! NeurIPS does not allow uploading the revised manuscript or giving external links during the discussion periods. We will upload the revised manuscript as a camera-ready version. Please let us know if you would like to discuss further the first question or any other aspect of our work. We greatly appreciate your effort in making our paper better!
> > >
> > > Sincerely,
> > >
> > > The authors

---

### Official Review · Reviewer_4gyW · 2023-07-09

**Soundness:** 4 excellent
**Presentation:** 4 excellent
**Contribution:** 4 excellent
**Rating:** 6
**Confidence:** 4

**Summary:**

The paper introduces a novel approach to ensembling multiple vision-language models (VLMs) for solving visual reasoning tasks. More specifically, the authors propose to use a language model (LM) to coordinate answers from various VLMs, which outperforms traditional ensemble approaches. Multiple experiments demonstrate the effectiveness of the proposed Cola approach.

**Strengths:**

1. The paper is well written, with great explanations of the proposed Cola approach and various figures and tables.

2. Using the LM to coordinate VLMs for visual reasoning tasks is novel and interesting.

**Weaknesses:**

The authors observed that long input (multiple VLMs) does not guarantee higher performance in Section 3.5. Specifically, Figure 4 shows the approach is vulnerable to the number of models. Cola models may also be affected by which VLMs are used.

**Questions:**

In Table 3, the authors implemented "Ensemble (average)" based on Equation (1). Was each $P_i(v, q)$ normalized before averaging them?

**Limitations:**

The authors discussed the limitations of the work in Section 5.

---

> ### Author Rebuttal · Authors · 2023-08-08
>
> We sincerely appreciate the reviewer's insightful critique and thoughtful suggestions for improving our work. In response, we have substantially revised the paper to better demonstrate the effectiveness of language models for coordinating multiple vision-language models.
>
> We humbly ask the reviewer to meticulously check our responses and the revisions. We greatly value your time, effort, and acknowledgment you have given to our work.
>
> **Q: The authors observed that long input (multiple VLMs) does not guarantee higher performance in Section 3.5. Specifically, Figure 4 shows the approach is vulnerable to the number of models.**
>
> Regarding the limitation on long context inputs, we agree this is an inherent challenge for LLMs. Longer text input is a significant challenge for LLM itself. It’s a interesting topic that should be investigated further and we have added references [1,2,3] to recent works that propose techniques to extend LLM ability to model long context, which can potentially enhance Cola in the future.
>
> [1] Chen, S., Wong, S., Chen, L., & Tian, Y. (2023). Extending context window of large language models via positional interpolation. *arXiv preprint arXiv:2306.15595*.
>
> [2] Mu, J., Li, X. L., & Goodman, N. (2023). Learning to compress prompts with gist tokens. *arXiv preprint arXiv:2304.08467*.
>
> [3] Bulatov, A., Kuratov, Y., & Burtsev, M. (2022). Recurrent memory transformer. *Advances in Neural Information Processing Systems*, *35*, 11079-11091.
>
> **Q: In Table 3, the authors implemented "Ensemble (average)" based on Equation (1). Was each $P_i(v, q)$ normalized before averaging them?**
>
> We also appreciate the clarification request on ensemble prediction averaging. To confirm, we did not normalize the per-model scores $P_i(v, q)$ before averaging and majority voting, since they are already on the same 0-1 scale. Normalization is not needed in this circumstance. Thank you for catching this detail - we have updated the text to clearly state that no normalization was applied.

---

> > ### Author Response · Authors · 2023-08-14
> > **Sincerely Looking Forward to Your Reply**
> >
> > Dear Reviewer,
> >
> > Your suggestions and comments have greatly helped polish our paper, regarding the long input and the ensemble baseline. We sincerely look forward to your reply to our response, and we are open to any discussion to improve our paper.
> >
> > Best wishes,
> >
> > The authors

---

> > > ### Comment · Reviewer_4gyW · 2023-08-20
> > >
> > > Thanks for the rebuttal. The authors addressed my concerns, and I will keep my original rating.

---

### Author Rebuttal · Authors · 2023-08-08

We greatly appreciate the reviewers' time and effort in providing thoughtful feedback on our work. We are pleased that the reviewers recognize the novelty of our Cola framework for coordinating multiple VLMs for visual reasoning. We also appreciate the suggestions to strengthen the paper through additional experiments and analysis.

First, we are deeply appreciative of the reviewer’s recognition of our paper by their comments on:

- Introduction of a novel Cola approach using a language model (LM) to effectively coordinate multiple visual language models (VLMs) for visual reasoning tasks;
- Comprehensive experiments that highlight Cola's notable performance across several datasets;
- Detailed analysis, inclusive of ablation studies and visual illustrations, confirming the coordinator LM's adeptness at integrating VLMs.

In response, we have conducted several new ablation studies which provide further insights into Cola:

- Rationales help explain Cola's reasoning steps but do not improve performance, confirming that the coordinator alone captures the necessary reasoning.
- Cola boosts the latest SOTA VLMs like InstructBLIP, showing general effectiveness across model families.
- Cola improves compositional reasoning on GQA and CLEVR, demonstrating broad applicability.

Additionally, we present a detailed analysis differentiating the behaviors of Cola-Zero and Cola-FT. References are provided to specifically address the requests of Reviewers 3ict and Yu6H.

Through these added experiments and analyses, we believe the paper better conveys the strengths of our approach. We are grateful to the reviewers for their feedback pushing us to strengthen the work, and we hope the revisions satisfactorily address the concerns raised. We look forward to continuing the discussion and thank the reviewers for their time and consideration.

---

### Author Response · Authors · 2023-08-18
**Looking Forward to Further Discussions**

Dear reviewers,

Thank you again for your valuable time and insightful comments! We sincerely look forward to your reply to our response to let us know if we have resolved your concerns, and we are open to any discussion to improve our paper.

Best regards，

The authors

---

### Decision · Program_Chairs · 2023-09-21

**Decision:**

Accept (poster)

**Comment:**

The paper combines the use of a LM with two VLMs to address visual question answering. The two VLMs are coordinated through the LM through careful prompting, an additional contribution is demonstrating the benefit of finetuning the LM on the task and demonstrating the effect of LMs at different scales. Reviewers assessed the work mostly positively but there were initial concerns on lack of experiments on other datasets such as GQA and CLEVR. Additional results were provided showing the effectiveness of the method across more diverse scenarios which was viewed positively by reviewers.

The paper is overall good and authors are encouraged to incorporate reviewer feedback and additional supporting experiments in the final version.